# Ocean acidification increases the sensitivity and variability of physiological responses of an intertidal limpet to thermal stress

Jie Wang[1], Bayden D. Russell[2], Meng-wen Ding[1], Yun-wei Dong[1,3]*

[1]State Key Laboratory of Marine Environmental Science, College of Ocean and Earth Sciences, Xiamen University, Xiamen, 361000, China

[2]The Swire Institute of Marine Science and School of Biological Sciences, The University of Hong Kong, Hong Kong SAR, 999077, China

[3]Fujian Collaborative Innovation Center for Exploitation and Utilization of Marine Biological Resources, Xiamen University, Xiamen 361102, China

*Corresponding to*: Yun-wei Dong (dongyw@xmu.edu.cn)

**Abstract.** Understanding physiological responses of organisms to warming and ocean acidification is the first step towards predicting the potential population- and community-level ecological impacts of these stressors. Increasingly, physiological plasticity is being recognized as important for organisms to adapt to the changing microclimates. Here, we evaluate the importance of physiological plasticity for coping with ocean acidification and elevated temperature, and its variability among individuals, of the intertidal limpet *Cellana toreuma* from the same population in Xiamen. Limpets were collected from shaded mid-intertidal rock surfaces. They were acclimated under combinations of different $pCO_2$ concentrations (400 ppm and 1000 ppm, corresponding to pH 8.1 and 7.8) and temperatures (20 °C and 24 °C) in a short-term period (7 days), with the control condition (20 °C and 400 ppm) representing the average annual temperature and present-day $pCO_2$ level at the collection site. Heart rates (as a proxy for metabolic performance) and expression of genes encoding inducible and constitutive heat-shock proteins (*hsp70* and *hsc70*) at different heat shock temperatures (26, 30, 34 and 38 °C) were measured. Hsp70 and Hsc70 play important roles in protecting cells from heat stresses, but have different expression patterns with Hsp70 significantly increased in expression during stress and Hsc70 constitutively expressed and only mildly induced during stress. Analysis of heart rate showed significantly higher temperature coefficients ($Q_{10}$ rates) for limpets at 20 °C than at 24 °C and post-acclimation thermal sensitivity of limpets at 400 ppm was lower than at 1000 ppm. Expression of *hsp70* linearly increased with the increasing heat-shock temperatures, with the largest slope occurring in limpets acclimated under a future scenario (24 °C and 1000 ppm $pCO_2$). These results suggested that limpets showed increased sensitivity and stress response under future conditions. Furthermore, the increased variation in physiological response under the future scenario indicated that some individuals have higher physiological plasticity to cope with these conditions. While short-term acclimation to reduced pH seawater decreases the ability of partial individuals against thermal stress, physiological plasticity and variability seem to be crucial in allowing some intertidal animals to survive in a rapidly changing environment.

## 1   Introduction

Benthic organisms living in the intertidal zone will be exposed to increasingly variable and extreme environmental conditions, such as temperature, oxygen and $CO_2$, due to climatic change (IPCC, 2013; Kwiatkowski et al., 2016). These highly fluctuating environmental variables can significantly affect the physiological performance of coastal species (Helmuth et al., 2006; Hofmann and Todgham, 2010; Somero, 2012; Widdicombe and Spicer, 2008). Therefore, understanding the interaction of multiple environmental stressors on the physiological performance is crucial for predicting the consequences of environmental change on ecosystems (Deutsch et al., 2015). For example, salinity fluctuations coupled with high temperatures during emersion can have both sub-lethal physiological effects and lethal effects on intertidal molluscs (Dong et al., 2014; Firth and Williams, 2009). Although ocean acidification can increase the growth of organisms in some cases (e.g. Gooding et al., 2009), there is increasing evidence that decreased pH exacerbates global warming, and interactions of ocean acidification and warming reduce resistance of an organism to environmental change (Munday et al., 2009) and subsequently affect population dynamics (Fabry et al., 2008; Hoegh-Guldberg et al., 2007; Kroeker et al., 2013; Rodolfo-Metalpa et al., 2011).

In the face of a changing environment, organisms can respond in three ways: exhibit shifts in distributional ranges (Parmesan and Yohe, 2003), develop adaptive changes (Hoffmann and Sgro, 2011), or perish (Fabricius et al., 2011). Prior to mortality or range-shifts, environmental changes can often drive physiological adaptation or the evolution of phenotypic plasticity (Chevin et al., 2010; Sanford and Kelly, 2011). Yet, warming and ocean acidification are not unidirectional, but rather combined with rapid fluctuations on daily to seasonal and decadal time-scales. Thus, the changing environment often does not provide clear signals to drive strong directional selection of traits, meaning that, usually, physiological

plasticity is the more important factor in acclimation to changing environmental conditions (Hoffmann
and Sgro, 2011; Pörtner et al., 2012; Somero et al., 2012). In a recent meta-analysis, Seebacher et al.
(2015) demonstrated that acclimation to higher temperatures decreased the sensitivity to increased
temperature in both freshwater and marine animals. While this response suggests that acclimation could
reduce the impact of warming on organisms, the responses were only tested for shifts in mean
temperature. Yet, organisms inhabiting variable environments, such as the intertidal zone, will be exposed
to increasing extremes in temperature concomitant with increasing $pCO_2$, or ocean acidification (OA),
in the future. While OA has been suggested to increase the sensitivity of organisms to warming (Byrne
and Przeslawski, 2013; Byrne, 2011; Kroeker et al., 2013), physiological plasticity and variation in
responses may provide the basis for populations to survive.
Physiological variation, or plasticity, within population is important for adapting to local
microclimate and for evolution (Dong et al., 2017; Oleksiak et al., 2002; Prosser, 1955). For example,
different color morphs of the gastropod *Littorina saxatilis* have enhanced physiological performance
which leads to increased survival under extreme conditions, indicating physiological differences may
provide a selective advantage for those color morphs under extremely fluctuating salinity and
temperature regime in estuaries (Sokolova and Berger, 2000). For the limpet *Cellana toreuma*, highly
variable expressions of genes related to stress responses and energy metabolism are important for
surviving the harsh environment on subtropical rocky shores (Dong et al., 2014).
Heart rate, as a measure of cardiac activity, is a useful indicator for indicating physiological response
to stress in molluscs (Dong and Williams, 2011; Xing et al., 2016). Animals exhibit a stable basal heart
rate under conditions which are not thermally stressful, and heart rate increases and reaches a peak
followed by a sudden decrease with temperature rising (Braby and Somero, 2006; Dong and Williams,
2011). The temperature at which a sharp discontinuity in slope occurs in an Arrhenius plot (i.e. Arrhenius
Breakpoint Temperature, ABT) can represent the limit of metabolic functioning of animals (Nickerson
et al., 1989; Somero, 2002). At the molecular level, expression of heat shock proteins (Hsps) and *hsp*
genes is induced above a certain temperature, reaches maximum and finally ceases in response to heat
shock (Han et al., 2013; Miller et al., 2009). Upregulation of Hsps and *hsp* genes is an energy-consuming
mechanism for defense against thermal stress (Somero et al., 2016). As a commonly used biomarker, the
Hsp70 multigenic family includes two proteins with divergent expression patterns (inducible Hsp70 and
constitutive Hsc70). The inducible Hsp70 significantly increases in expression when animals are exposed
to stressors and plays a role in maintaining protein stability (Feder and Hofmann, 1999); on the other
hand, the constitutive Hsc70, which is transcribed continuously and may be mildly induced during stress,
takes part in folding and repairing of denatured proteins (Dong et al., 2015) and plays a role in the
formation of mitotic structures (Sconzo et al., 1999). Some studies have shown coordinated heart rate
and expression of genes encoding to Hsps in response to elevated temperate (Han et al., 2013; Prusina et
al., 2014). However, little is known about the patterns of heart rate and expression of *hsp* genes for coping
with combined warming and ocean acidification.

The limpet *C. toreuma* is a keystone species on rocky shores in the western Pacific (Dong et al.,

2012), occupying the mid-low intertidal zones (Morton and Morton, 1983). This species is a gonochoric
and broadcast spawner, whose embryos develop into planktonic trocophore larvae and later into juvenile
veligers before becoming fully grown adults (Ruppert et al., 2004). As a common calcifier inhabiting
coastal ecosystems, *C. toreuma* plays an important ecological role in food chains, gazing on biofilm and
being an important food source for other species (e.g. crabs, sea birds and sea stars). Therefore, this
species is a key organism for studying the relationship between physiological response to thermal stress
and ocean acidification in highly variable environment on the shore.

Under the impact of subtropical high pressure systems, Xiamen (118°14′ E, 24°42′ N) is one of the

hottest areas in China (Dong et al. 2017). The coastal seawater of this area is experiencing rapid
temperature rise and acidification (Bao and Ren, 2014). The sea surface temperature (SST) in Xiamen
coastal water has increased a total of 1 °C since 1960, and is rising at a mean annual rate of 0.02 °C (Yan
et al., 2016). The annual pH values of seawater in Xiamen Bay have declined by 0.2 pH units from 8.05
in 1986 to 7.85 in 2012, a trend which is predicted to continue based on simulations (Cai et al., 2016).

Here, we investigated the importance of physiological plasticity (based on the measurement of post-

acclimation temperature sensitivity; see Seebacher et al., 2015) and variability (based on coefficient of
variation) for *C. toreuma* to cope with ocean acidification and elevated temperatures by quantifying heart
rates (as a proxy of metabolic performance) and expression of genes encoding inducible and constitutive
heat-shock proteins (Hsp70 and Hsc70) after short-term acclimation in different $p\text{CO}_2$ concentrations
(400 ppm and 1000 ppm) and temperatures (20 °C and 24 °C). We hypothesize that (1) limpets will show
increased thermal sensitivity of metabolism and stress responses under elevated $p\text{CO}_2$ and temperatures;
(2) short-term acclimation at high temperature and $p\text{CO}_2$ will cause higher inter-individual physiological
variation. Our study, by measuring both heart rate and heat shock protein gene expression, provides novel
information concerning the combined effects of increased temperature and $p\text{CO}_2$ on stress response,
energy consumption and physiological plasticity in intertidal invertebrates, potentially providing
predications of the ecological impacts of the future environmental changes.

## 2    Material and Methods

### 2.1  Limpet collection and experiment treatments

Samples were collected from shaded rock surfaces at mid-tidal level in Xiamen on a falling high

tide in July (*in situ* temperature: $30.8 \pm 0.8$ °C). The sampling ensured that all limpets have similar

thermal history, given the possible impacts from microclimate (Dong et al., 2017; Lathlean and Seuront,

2014). They were transported to the State Key Laboratory of Marine Environmental Science, Xiamen

University, China within 2 h. Limpets were firstly allowed to recover at 20 °C for 3 d with a tidal cycle

of approximately 6 h immersion and 6 h emersion. These limpets were randomly allocated into one of

four treatments and temporally acclimated in different $p$CO$_2$ concentrations and temperatures (LTLC,

low temperature and low CO$_2$, 20 °C + 400 ppm, as a control treatment; LTHC, low temperature and

high CO$_2$, 20 °C + 1000 ppm; HTLC, high temperature and low CO$_2$, 24 °C + 400 ppm; HTHC, high

temperature and high CO$_2$, 24 °C + 1000 ppm) for 7 d in climate chambers (RXZ280A, Jiangnan

Instrument Company, Ningbo, China), which control both the $p$CO2 concentration and temperature under

the same relative humidity and light intensity conditions. In each acclimation treatment, approximately

100 limpets were randomly allocated in ~ 30 containers (3 individuals in each container), to simulate

field densities of ~ 1 limpet per 10 cm$^2$. Control conditions (20 °C, 400 ppm) represent the average annual

temperature and ambient $p$CO$_2$ (~ 390 ppm) at the collection site, with high temperature (24 °C) and

$p$CO$_2$ (1000 ppm) representing the average global increase (4 °C, 600 ppm) predicted for 2100 by the

Intergovernmental Panel on Climate Change (IPCC, 2007).

Animals were kept in a simulated tidal cycle with 6 h aerial exposure and 6 h seawater immersion.

Seawater was pre-bubbled with air containing the corresponding $p$CO$_2$ concentrations in advance. pH

was measured before and after the acclimation in seawater each time with PB-10 pH meter (Sartorius

Instruments, Germany) calibrated with National Institute of Standards and Technology standard pH
solutions (NIST, USA). Total dissolved inorganic carbon (DIC) was measured before and after the
acclimation in seawater each time using a dissolved inorganic carbon analyzer (As-C3, Apollo SciTech,
Colorado, USA), using a Li-Cor® non-dispersive infrared detector (Li-6252) with a precision of 0.1%
(Cai, 2003). Seawater carbonate chemistry parameters were estimated based on the measured values of
pH, DIC, temperature and salinity with the software CO2Calc v4.0.9 (Robbins et al., 2010). For CO2Calc
settings, the NBS scale was applied as the pH scale, and the $CO_2$ constant, the $KHSO_4^-$ constant and the
total Boron was set from Millero et al. (2006), Dickson et al. (1990) and Lee et al. (2010) respectively.
The information of the measured and calculated seawater chemistry parameters is summarized (Table 1).

After a 7-day acclimation period (crossed $p$CO$_2$ × Temperature treatments, above), the heat-shock

treatments were carried out to simulate the gradual temperature exposure of limpets in the field as
described in Denny et al. (2006) (Fig. A1). For each heat-shock treatment, 10 limpets were randomly
selected from each of four acclimation conditions (40 indiv. total) and transferred to artificial rocks (Fig.
A2), with individuals from LTLC and LTHC on one rock and individuals from HTLC and HTHC on
another rock. The artificial rocks were separately placed in 20 °C water baths and 24 °C water baths, and
heated at a rate of 6 °C per hour that simulated emersion in the natural condition at the collection site
(Han et al., 2013) to the designated temperatures (26, 30, 34 and 38 °C). After achieving the target
temperature, the temperature was maintained for the allotted time, and then decreased to the acclimation
temperature (20 or 24 °C) at a rate of 6 °C per hour, for a total exposure time of 7 h. Individuals from all
four acclimation conditions (n = 10 indiv. per treatment) were randomly selected, transferred to artificial
rocks and aerially exposed at 20 or 24 °C for 7 h, as non-heated control samples. After recovery at 20 or
24 °C seawater for 1 h, limpets were immediately collected and stored at -80 °C for gene expression
analysis.

**2.2  Cardiac performance measurement**
The cardiac performance of limpets was recorded during whole heating processes from the
acclimated temperature (20 or 24 °C) to the temperature where the heart stopped beating (n = 9-11 indiv.
per acclimation treatment). Each limpet was placed in a separate container during the measurement. The
containers were immersed in water baths, allowing the temperature in the container to be increased at a
rate of 6 °C per hour that simulated emersion in the natural environment. Heart rates were measured
using a non-invasive method (Chelazzi et al., 2001; Dong and Williams, 2011). The heartbeat was
detected by means of an infrared sensor fixed with Blue-Tac (Bostik, Staffordshire, UK) on the limpet
shell at a position above the heart. Variations in the light-dependent current produced by the heartbeat
were amplified, filtered and recorded using an infrared signal amplifier (AMP03, Newshift, Leiria,
Portugal) and Powerlab AD converter (4/30, ADInstruments, March-Hugstetten, Germany). Data were
viewed and analyzed using Lab Chart (version 7.0).
For determining the Arrhenius Breakpoint Temperature (ABT) of heart rate, discontinuities in the
slopes of heart rate with temperature were calculated from intersections of fitted 2-phase regressions
based on the minimum sum of squares using SigmaPlot 12.5 (SSPS Inc., Point Richmond, CA, USA) as
described by Giomi and Pörtner (2013).

## 2.3  Quantifying genes expression

Limpets were firstly taken out from – 80 °C; the foot muscle was cut off immediately using RNA-free scissors (180 °C before using); the muscle (~ 50 mg) was cut into pieces in a 1.5 ml EP tube containing RNA lysis buffer provided by Eastep reagent kit (Promega, USA); total RNA was isolated using Eastep reagent kit (Promega, USA). The first strand of cDNA was synthesized using total RNA as a template. Reverse transcriptase (RT) reactions were performed using a PrimeScript RT reagent kit with gDNA Eraser (Takara, Shiga, Japan).

The levels of mRNA of genes encoding two heats hock proteins, inducible heat-shock protein 70 (*hsp70*) and constitutive heat shock protein 70 (*hsc70*), were measured using real-time quantitative PCRs in CFX96™ Real-Time System (Bio-Rad Laboratories, Inc., Hercules CA, USA) followed the methods described by Han et al. (2013) with specific primers (Table A1). For normalizing expression of genes, we examined expression of *18S ribosomal RNA*, *β-actin*, *β-tubulin* genes, which typically have relatively stable expression levels. The expression stability of these housekeeping genes was evaluated using the GeNorm Algorithm (Primer Design, Ltd., Southampton University, Highfield Campus, Southampton Hants, UK) as described by Etschmann et al. (2006). Based on the expression stability measures (M values), all the three genes were selected as the reference genes for normalizing the level of expression of stress-induced genes. All samples were measured in triplicates. Ct (dR) values were analyzed using the CFX Manager™ Software Version 3.0 (Bio-Rad). The expression of *hsp70* and *hsc70* was determined relative to the value of *18S*, *β-actin* and *β-tublin* from a reference individual.

**2.4  Statistical analysis**

The general additive mixed model (GAMM) was used to compare thermal sensitivities of heart rate

among limpets acclimated at different temperatures (20 or 24 °C) and $CO_2$ concentrations (400 or 1000
ppm). Analyses were conducted with the *mgcv* (Wood, 2004) and *nlme* (Pinheiro et al., 2013) libraries in
R Version 3.0 (R Core Team, 2014). The generalized additive model (GAM), describing heart rate as a
function of temperature, was used to test for how heart rates of limpets from each treatment deviated
from those of limpets from control conditions (20 °C, 400 ppm) (Angilletta et al., 2013).

Thermal sensitivity is the change in a physiological rate function reacting to a rapid change in

environmental temperature within the same acclimation set temperature (Fig. A3, modified from
Seebacher et al. (2015)). In the present study, thermal sensitivity was determined in the temperature
coefficient ($Q_{10}$) values of heart rate. $Q_{10}$ was calculated using heart-rate data from the temperature at
which the experiment started ($T_1 = 24$ °C) to the temperature to which temperature increased 10 °C ($T_2$
$= 33$ °C) with Eq. (1):
$$Q_{10} = \left(\frac{R_2}{R_1}\right)^{\frac{10}{T_2 - T_1}} \tag{1}$$
where R is the heart rate ($R_1$ and $R_2$ are the heart rate at $T_1$ and $T_2$ respectively), and T is the temperature
(Kelvin) (Fig. A3, modified from Seebacher et al. (2015)). The differences in $Q_{10}$ among the four
acclimation conditions with different $CO_2$ concentrations (400 ppm vs. 1000 ppm) and temperatures
(20 °C vs. 24 °C) were analyzed using two-way ANOVA with Duncan's *post hoc* analysis using the SPSS
20.0 for Windows statistical package (IBM SPSS Statistics, Chicago, USA). Post-acclimation thermal
sensitivity of limpets in different $CO_2$ concentrations were calculated as described by Seebacher et al.
(2015). In each $CO_2$ concentration (400 ppm or 1000 ppm), the post-acclimation $Q_{10}$ values were
calculated using the same equation as shown above, but $R_2$ was the average heart rate of the warm-
acclimated limpets at the acclimated temperature ($T_2$ = 24 °C), and $R_1$ was the average heart rate of cold-
acclimated limpets at $T_1$ = 20 °C (Fig. A3, modified from Seebacher et al. (2015)). It is worth noting that
post-acclimation thermal sensitivity should be considered with caution, as in the present study the
acclimation period (7 days) may not have been sufficient for full acclimation to altered conditions.
The differences in levels of *hsp70* and *hsc70* among different heat shock temperatures within a same
acclimation condition were analyzed using one-way ANOVA with Duncan's *post hoc* analysis. The
relationships between heat shock temperature and log-transformed gene expression (*hsp70* and *hsc70*)
were fitted using linear regressions and the differences in slopes of the linear regressions were analyzed
using Analysis of Covariance (ANCOVA).
The coefficient of variation (CV) of ABT, $Q_{10}$ and *hsc70* mRNA expression at 38 °C were
calculated for each acclimation condition. The CV is the variance in a sample divided by the mean of
that sample, providing a method to compare the variation within a sample relative to the mean. It is
generally accepted that higher CV demonstrates that there is greater variation among individuals within
one treatment than another (Reed et al., 2002).

**3   Results**
**3.1  Cardiac performance**
The maximal heart rate was ~ 30 % higher in limpets acclimated to control conditions (20 °C, 400
ppm) than the other treatments (Fig. 1 and Table A2). The ABTs of limpets showed a trend to be reduced
for high temperature treatments (mean ± SD: LTLC, 38.9 ± 2.9 °C; HTLC, 38.2 ± 1.8 °C; LTHC, 40.0 ±
3.3 °C; HTHC, 37.7 ± 2.3 °C) (Fig. A4). Temperature (Two-way ANOVA, $F_{1, 35}$ = 3.375, P = 0.075) and
$p$CO$_2$ (Two-way ANOVA, $F_{1, 35} = 0.118$, P = 0.733) both had non-significant effects on ABTs, and there
was a non-significant interaction between temperature and $p$CO$_2$ (Two-way ANOVA, $F_{1, 35} = 0.908$, P =
0.347) (Table A3; Fig. A4).

Temperature coefficients ($Q_{10}$ rates) were higher for limpets acclimated at 20 °C than at 24 °C (Two-

way ANOVA, $F_{1, 35} = 5.878$, P = 0.02), but there was no significant difference for acclimation to different
$p$CO$_2$ concentrations (Two-way ANOVA, $F_{1, 35} = 1.332$, P > 0.05) and for the interaction between
temperature and $p$CO$_2$ (Two-way ANOVA, $F_{1, 35} = 0.1135$, P > 0.05) (Table A3; Fig. 2). The post-
acclimation thermal sensitivity of limpets acclimated at low CO$_2$ (2.12) was lower than that of limpets at
high CO$_2$ (2.95) (Fig. 2).

The coefficients of variations (CV) of ABT in the four different acclimation conditions were

different (Table 2). After low temperature and high CO$_2$ acclimation (LTHC, 8.22%), CV of ABT was
higher than those in the other three conditions (LTLC, 7.34% and HTLC, 4.48%, HTHC, 6.08%). CV of
$Q_{10}$ under LTHC condition was the highest in all the four acclimation conditions (Table 2).

**3.2 Gene expression**

Levels of *hsp70* mRNA (log-transformed) linearly increased with the increasing heat-shock

temperatures (Fig. 3). ANCOVA analysis showed that the slopes of the linear regressions were
significantly different among different acclimation conditions ($F_{4, 189} = 42.62$, P < 0.001), and the slope
of HTHC condition was higher than those of the other three acclimation conditions. Thus, the rate of
increase in production of *hsp70* mRNA in response to warming was greater at the elevated CO$_2$
concentration.
The responses of *hsc70* mRNA to heat shock were divergent among the four acclimation conditions
(Fig. 4). For HTHC limpets, there were no significant differences among different heat shock
temperatures ($F_{4, 42} = 2.11$, $P = 0.096$). For LTLC, LTHC and HTLC limpets, levels of *hsc70* mRNA after
being heat-shocked at 38°C were higher than corresponding levels of *hsc70* mRNA at 20 °C or 24 °C
(Duncan's *post hoc* analysis, $F_{4, 42} = 4.389$, $P = 0.005$; $F_{4, 44} = 8.521$, $P < 0.0001$; $F_{4, 42} = 5.713$, $P = 0.001$).
The coefficients of variation of *hsc* mRNA after heat shock of 38°C were different among different
acclimation conditions: HTHC (90.36%) > LTHC (80.44%) ≈ HCLT (80.12%) > LCLT (56.20%) (Table

2).


**4    Discussion**
Short-term acclimation at elevated temperature and $p$CO$_2$ can increase physiological sensitivity of
limpets to thermal stress. The higher thermal sensitivity of limpets acclimated to 1000 ppm indicates that
the resilience of limpets to thermal stress associated with warming will be compromised under future
ocean acidification. This prediction is contrary to the general thought that intertidal ectotherms, such as
limpets and other gastropods, will demonstrate high tolerance to thermal stress because they are adapted
to an extreme thermal environment. For example, the operative temperatures, which *C. toreuma* suffers
in the field, frequently exceed 40 °C in summer along Asian coastlines and the limpet can survive at
temperatures in excess of 45 °C (Dong et al., 2015). Our data show, however, that ocean acidification
will lead to increased sensitivity to changes to future thermal regimes, indicating a synergistic negative
effect. The change in the metabolic partitioning in individuals could ultimately lead to a decline in fitness
and population-level responses in the future.
Increased temperature and CO$_2$ elevated the sensitivity of heat shock responses to thermal stress. The
expression of inducible *hsp70* mRNA steadily increased from 20°C to 38°C for individuals across all
experimental treatments. However, rates of upregulation of *hsp70* mRNA in limpets acclimated at high
temperature and high $CO_2$ (HTHC) were significantly higher than those of limpets acclimated at the other
three acclimation conditions. As a molecular chaperon, Hsp70 protein plays crucial roles in maintaining
protein stability with the expense of a large amount of energy (Feder and Hofmann, 1999; Tomanek and
Sanford, 2003). By comparing the expression patterns of Hsp70 of different *Chlorostoma* species
(formerly *Tegula*) that have distinct vertical distribution, Tomanek and Somero (1999, 2000) found that
there existed interspecific difference in the frequency of the induction of Hsp70 synthesis and
interspecific divergence of the time-course of Hsp70 synthesis. These studies from genus *Chlorostoma*
suggested that species that live higher in the intertidal zone spend more energy for proteostasis and restore
proteostasis to cope with a second consecutive day of high temperatures (Somero et al., 2016). Usually,
the expression of Hsp70 of less thermal-tolerant species is more sensitive to increases in temperature
(limpet *Lottia*, Dong et al., 2008; snail *Chlorostoma*, Tomanek, 2002), and the rapid upregulation of
*hsp70* mRNA in limpets exposed to future conditions potentially represents a high sensitivity of limpets
to thermal stress in the face of ocean acidification. Due to the expensive energy consumption during the
synthesis and function of *hsp70*, the more rapid upregulation of *hsp70* mRNA in these limpets also
indicates more energy was allocated into cellular homeostasis, which then can affect the growth and
reproduction of limpets.

The expression patterns of constitutive *hsc70* mRNA were different among limpets acclimated at the

four acclimation conditions. Hsc70 is constitutively expressed and is a molecular chaperone involved in
the *in vivo* folding and repair of denatured proteins (Dong et al., 2015). Although *hsp70* and *hsc70* contain
similar promoter regions, there are differential expressions to a given stimulus between them (Hansen et
al., 1991). Some studies showed that thermal stress could significantly induce the up-regulation of both
$hsc70$ gene and Hsc70 protein in the killifish *Fundulus heteroclitus* (Fangue et al., 2006), the shrimp
*Penaeus monodon* (Chuang et al., 2007), and the coral *Veretillum cynomorium* (Teixeira et al., 2013). In
the present study, for limpets acclimated under HTLC and LTHC (i.e. only temperature or $CO_2$ condition
changed in comparison with the LTLC treatment), there was significant upregulation of $hsc70$ mRNA
when the heat shock temperatures were beyond 30 °C. However, the expression of $hsc70$ mRNA showed
no significant difference among different heat-shock temperatures under predicated future environmental
conditions (HTHC: 24 °C and 1000 ppm). These results indicate that the upregulation of $hsc70$ mRNA
in response to heat shock represents an increasing capability for coping with the enhanced protein
denaturation and more energy allocated into the somatic maintenance after being exposed to either
warming or high $CO_2$ environment. The insignificant upregulation of $hsc70$ in response to thermal stress
indicates that limpets acclimated under HTHC may employ a "preparative defense" strategy (Dong et al.,
2008) to maintain high constitutive levels of $hsc70$ as a mechanism to copy with unpredictable heat stress.
However, the absence of significant upregulation of $hsc70$ mRNA in limpets acclimated to future
conditions (warming and elevated $CO_2$) might also be attributed to the very high variation of gene
expression at 38°C (CV, 90.36 %). In the context of future conditions, multiple environmental stressors
can induce diverse physiological responses among different individuals, which might be an evolutionary
adaptation to the harsh environment on the shore.
Variation and plasticity in both physiological and molecular responses to thermal stress are not only
important for coping with future environmental changes but also underpin evolutionary and adaptive
changes through selective pressures (Franks and Hoffmann, 2012). In the present study, the coefficients
of variation in physiological responses of limpets acclimated to simulated future conditions, including
ABT, $Q_{10}$ and *hsc70* mRNA, were higher than those in the other three acclimation conditions. Crucially,
this means that a subset of individuals in our experimental population might be more physiologically
pre-adapted to cope with heat shock. Once acclimated to future climate change scenario (warming and
ocean acidification), this variation in physiological performance increased, indicating that in a harsher
environment the physiological plasticity of some individuals allows them to modify their physiological
tolerance limits and increase chances for survival and reproduction (Williams et al., 2008). Under high
selective pressure, these individuals would form the basis for future generations while less plastic
individuals would be removed from populations. However, differences among the coefficients of
variation need to be interpreted with caution, as multiple factors can cause this type of variation,
including the variable environmental history of individuals despite a 7-day acclimation, competition
among individuals during the acclimation period, or the sample size (around 10 limpets per treatment).
Intertidal limpets may experience two sorts of stressful temperature exposures in the field,
abrupt or gradual exposure (Denny et al., 2006). The present study showed the upregulation of *hsp70*
and *hsc70* expression in *C. toreuma* under gradual exposure. Similar expression patterns have been also
observed in Hsp70 under gradual thermal exposure in other intertidal limpets (Dong et al., 2008; Miller
et al., 2009). Importantly, the gradual experimental change in thermal environment used here mimics
conditions that most intertidal species experience in the field and is important for predicting how animals
will resolve prolonged aerial exposure during low tide. Conversely, experimentally simulating abrupt
thermal change helps us understand physiological responses to some extreme conditions, such as heat
wave (upregulation of *hsp70* in intertidal limpets, Prusina et al., 2014). Therefore, future work combining
both abrupt and gradual exposure may offer insight into how intertidal species respond to climate change
and extreme weather events in the future. Further, since our findings are based on static experimental
conditions, the results should be treated with caution when we predict the response of an organism to
future climate change in the highly variable natural environment. Therefore, future studies with long-
term acclimation, larger sample size, and variable treatment conditions are recommended in order to
validate our findings.
In conclusion, the resilience of intertidal limpets to thermal stress is weakened after exposure to
predicted future conditions for a short-term acclimation period (7 d). Yet, the combination of elevated
temperature and $CO_2$ concentration prompted divergence of physiological and molecular responses.
These results suggest that while organisms may be able to protect themselves from the damaging effects
of thermal stress in the short-term, changes to multiple environmental conditions may drive population-
level responses through physiological responses (e.g. Giomi et al., 2016). Further, the increased variation
in responses, and the observation that some individuals were more capable to physiologically cope with
the conditions, may be associated with intergenerational adaptation, but this speculation needs further
evidence. As the "weaker" individuals are lost, the offspring in the next generation will be better
physiologically adapted to warming under high-$CO_2$ conditions. Therefore, while elevated $CO_2$ and the
associated ocean acidification decrease the ability of many individuals to respond to thermal stress, it
appears that physiological plasticity and variability could be adaptive mechanisms in at least some
populations of intertidal organisms. Our research underlines the importance of physiological plasticity
and variability for coastal species coping with warming and ocean acidification.

**Authors' contributions**
B.D.R and Y.-W.D. designed experiments. W.J. and M.-W.D. conducted experiments. Y.-W.D., B.D.R,
W.J. and M.-W.D. performed analyses. The manuscript was co-written by Y.-W.D., W.J. and M.-W.D.,
and revised by B.D.R.

**Competing interests**
The authors declare no conflict of interests.

**Acknowledgements**
This work was supported by grants from National Natural Science Foundation of China (41476115,
41776135), Nature Science funds for Distinguished Young Scholars of Fujian Province, China
(2017J07003), Program for New Century Excellent Talents of Ministry of Education, China, and the
State Key laboratory of Marine Environmental Science visiting fellowship to B.D.R.

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

**Table 1.** Measured and calculated seawater carbonate chemistry variables of each acclimation treatment during the
experimental period[1]

|  | 20 °C & 400 ppm | 24 °C & 400 ppm | 20 °C & 1000 ppm | 24 °C & 1000 ppm |
|---|---|---|---|---|
| Temperature (°C) | 20.94±0.88 | 24.84±0.87 | 20.59±0.91 | 25.01±0.67 |
| Salinity (psu) | 27.89±0.88 | 27.96±0.75 | 28.18±0.75 | 27.79±0.58 |
| $A_T$ (μmol/kg) | 2082.70±191.28 | 2083.016±190.58 | 2081.19±165.93 | 2083.29±163.58 |
| $C_T$ (μmol/kg) | 1910.57±174.42 | 1910.57±174.42 | 1992.76±157.22 | 1992.15±149.76 |
| $p$CO$_2$ (μtam) | 562.18±83.20 | 561.81±83.04 | 1008.66±113.41 | 992.36±47.04 |
| pH (NBS scale) | 8.05±0.05 | 8.05±0.05 | 7.82±0.04 | 7.83±0.04 |
| $CO_3^{2-}$ (μmol/kg) | 130.50±21.25 | 130.64±20.85 | 81.64±11.76 | 83.42±11.95 |
| $\Omega_{cal}$ | 3.31±0.55 | 3.32±0.54 | 2.07±0.30 | 2.12±0.30 |

[1]Seawater temperature, salinity, pH and total dissolved inorganic carbon ($C_T$) were monitored every 6 h. Total
alkalinity ($A_T$), $p$CO$_2$, $CO_3^{2-}$ and $\Omega_{cal}$ were calculated using CO2SYS software. Results were pooled and averaged
over sampling times. Values are given as mean ± SD.

**Table 2.** Coefficients of variation (%) of Arrhenius Breakpoint Temperature (ABT), temperature coefficients ($Q_{10}$)
and *hsc70* mRNA expression at 38 °C[1,2]

| Temperature | $CO_2$ | ABT | $Q_{10}$ | *hsc70* mRNA |
|---|---|---|---|---|
| 20 | 400 | 7.34 | 10.23 | 56.20 |
| | 1000 | 8.22 | 15.08 | 80.44 |
| 24 | 400 | 4.48 | 10.08 | 80.12 |
| | 1000 | 6.08 | 11.82 | 90.36 |

[1]Temperature coefficients ($Q_{10}$) were calculated using heart rate from 24 to 33 °C.
[2]After acclimated at different $CO_2$ and temperature for one week, limpets (n = 8-10) from each acclimation treatment
were randomly selected and heat shocked at designated temperatures. Levels of *hsc70* mRNA at 38 °C in different
acclimation treatments were used for calculating coefficients of variation.


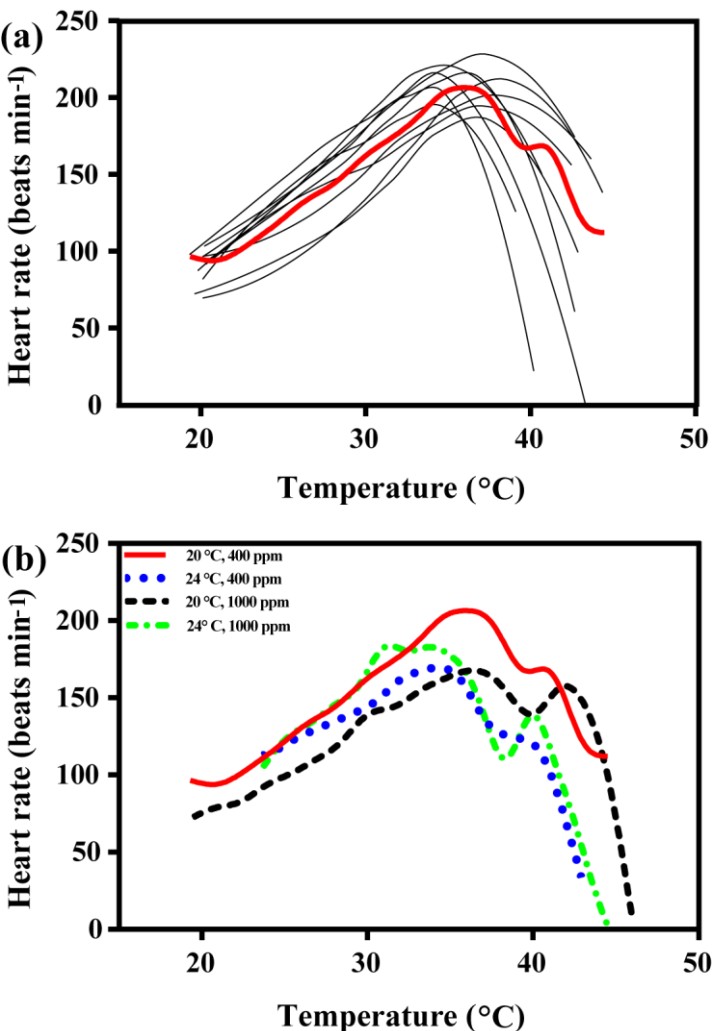



**Figure 1.** (a) Heart rates of all limpets acclimated to 20 °C and 400ppm, presented as an example of heart rate

calculation for limpets in all treatments. The black lines correspond to smoothed fits (using the loess algorithm) of

heart rates for each of the individual limpets. The red line represents the most likely general additive mixed model

(GAMM) to depict the trajectory of heart rates for limpets with increasing temperature; (b) GAMM lines of limpets

acclimated at the different experimental temperature and $CO_2$ conditions.

597

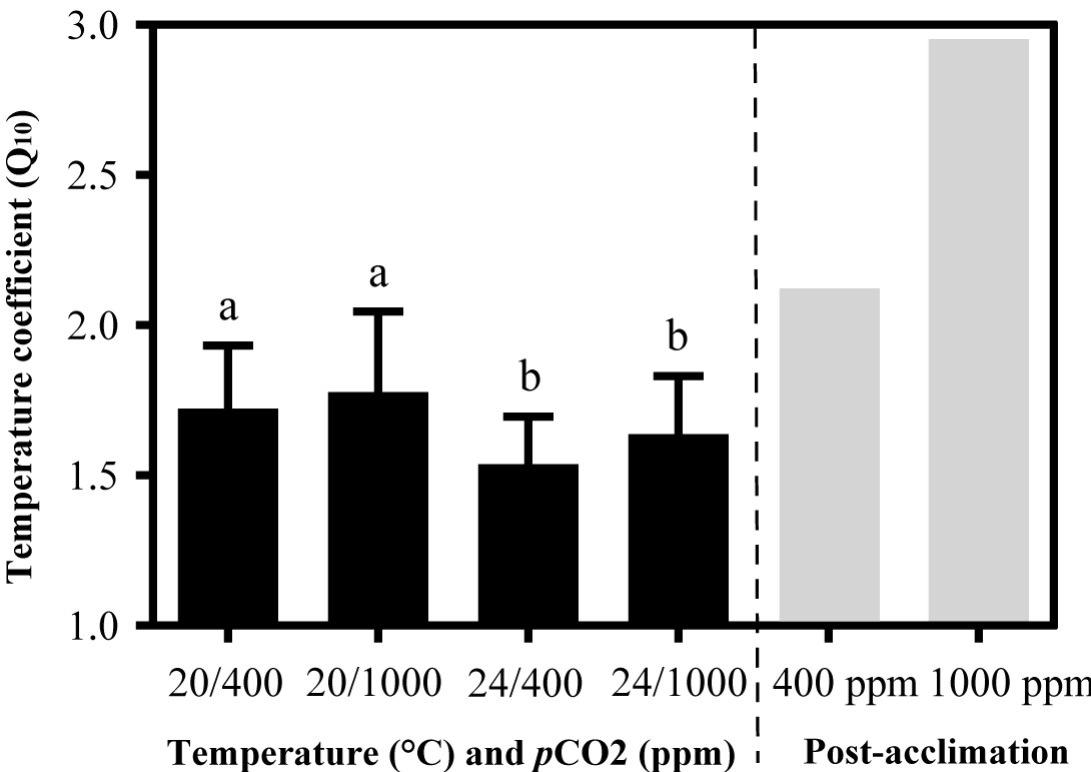

598

599

**Figure 2.** Temperature coefficients ($Q_{10}$) of limpets acclimated at different temperatures (20 or 24 °C) and $CO_2$

concentrations (400 or 1000 ppm). The temperature coefficient ($Q_{10}$) values were calculated for all limpets using

heart rate data from 24 to 33°C. Post-acclimation temperature sensitivity was calculated between individuals

acclimated at 20 and 24°C (grey bars; *sensu* Seebacher et al., 2015) for each $CO_2$ concentration, where higher thermal

sensitivity indicates less acclimation to thermal stress. The calculation of post-acclimation $Q_{10}$ is done for the mean

response of all individuals as the same individual are not used at each acclimation temperature. Therefore, it is not

possible to calculate an estimate of variation or error for post-acclimation $Q_{10}$. Different letters represent significant

differences in the $Q_{10}$ among different acclimation treatments.

608

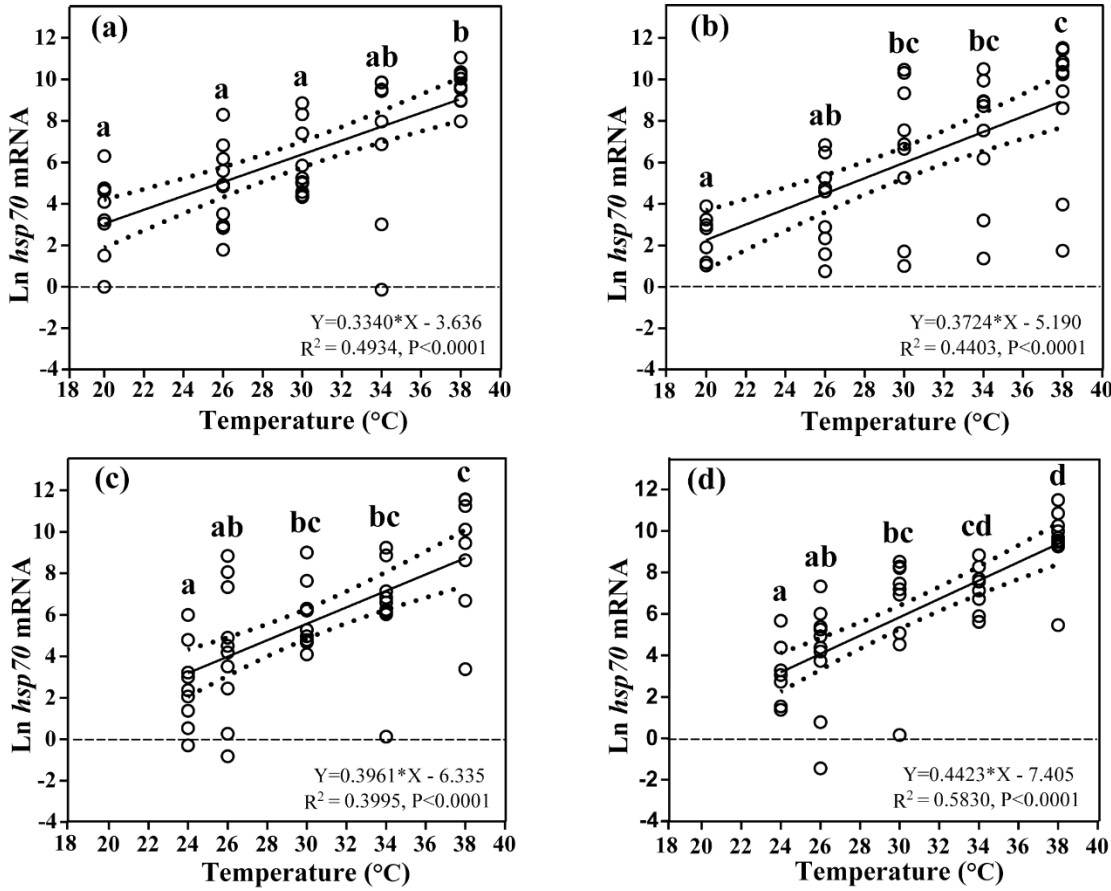

**Figure 3.** Effects of heat-shock temperature on the expression of *hsp70* mRNA in limpets acclimated at (a) 20°C

and 400 ppm, (b) 20°C and 1000 ppm, (c) 24°C and 400 ppm, and (d) 24°C and 1000 ppm. The relationship between

heat-shock temperature and log-transformed gene expression of *hsp70* was fitted using linear regressions with 95%

confidence intervals (dashed lines). Different letters represent significant differences in the level of *hsp70* mRNA

among different heat-shock temperatures.

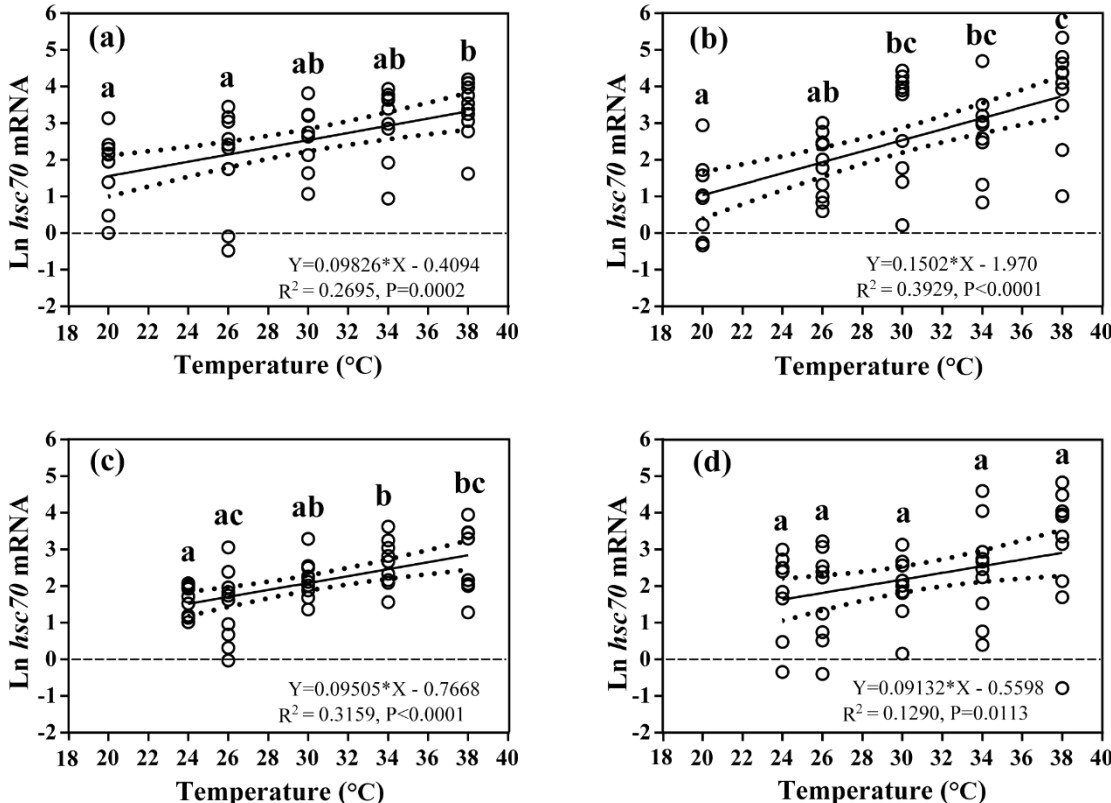

**Figure 4.** Effects of heat-shock temperature on the expression of *hsc70* mRNA in limpets acclimated at (a) 20°C and

400 ppm, (b) 20°C and 1000 ppm, (c) 24°C and 400 ppm, and (d) 24°C and 1000 ppm. The relationship between

heat-shock temperature and log-transformed gene expression of *hsc70* was fitted using linear regressions with 95%

confidence intervals (dahs lines). Different letters represent significant differences in the level of *hsc70* mRNA

among different heat-shock temperatures.

 **Appendix:**

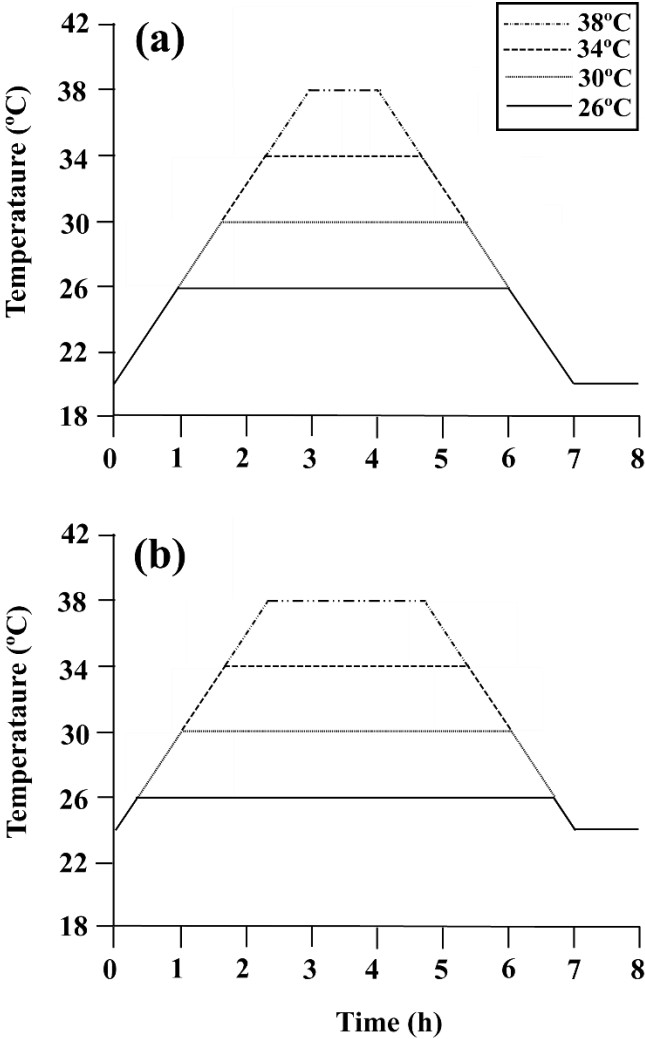

 **Figure A1.** Diagram of the heating protocol for (a) limpets acclimated at 20 °C and (b) limpets acclimated at 24 °C.

 Limpets were heated at a rate of 6°C per hour from acclimation temperatures (20 or 24 °C) to designated temperatures

 (26, 30, 34 and 38 °C) for simulating a natural heating rate in summer. After achieving the target temperature, the

 temperature was held at the designated level for the allotted time, and then decreased to acclimated temperatures (20

 or 24 °C) at a rate of 6 °C per hour, for a total exposure time of 7 h. After recovery in 20 or 24 °C seawater for 1 h,

 limpets (n = 8-10) in each treatment were immediately collected and stored at -80 °C for gene expression

 measurement.

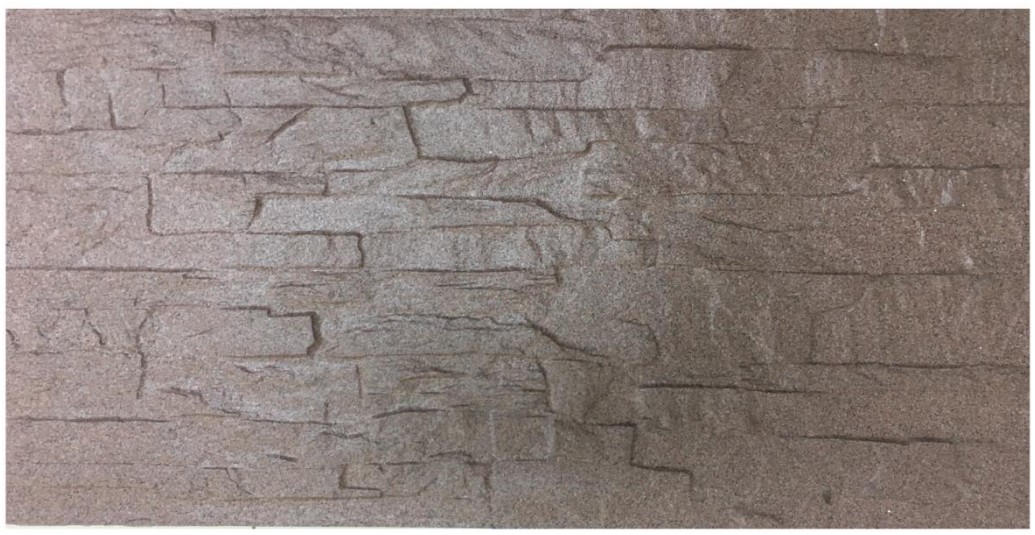


**Figure A2.** The photo of artificial rock (60 cm length × 30 cm width). Limpets were placed on artificial rock and
heated to the designated temperate.


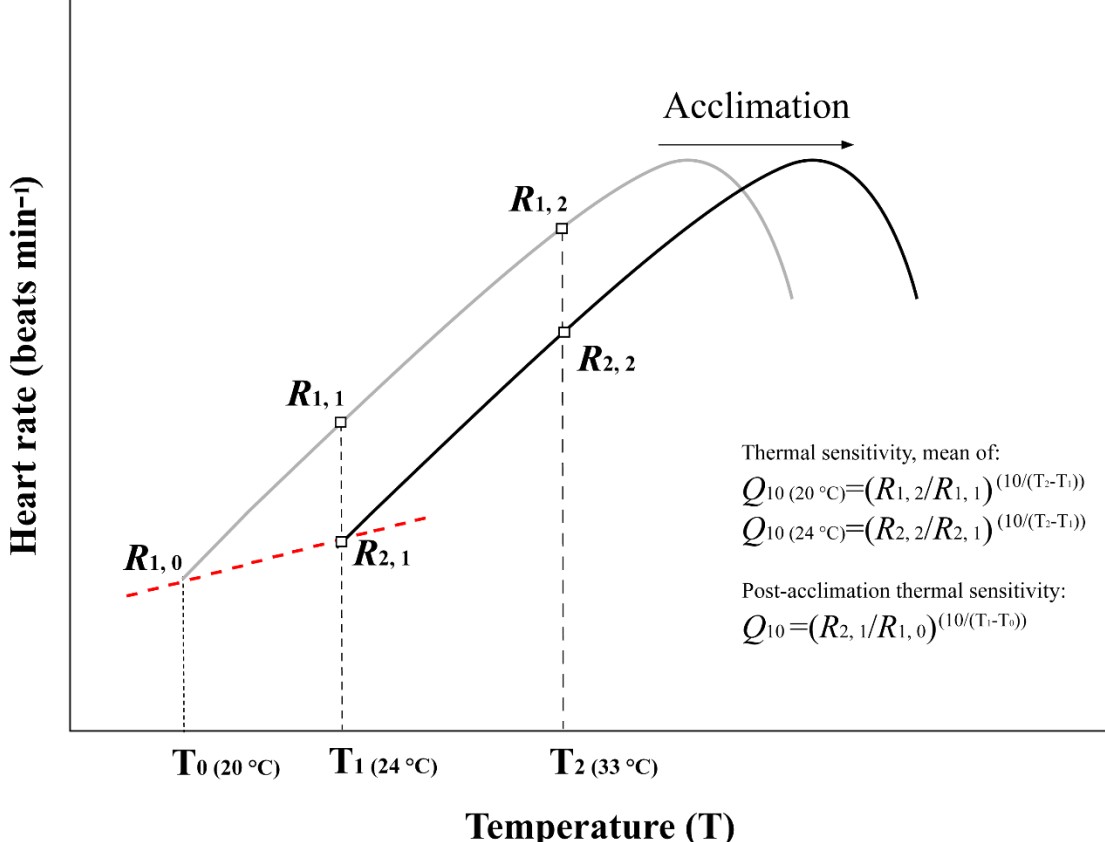



**Figure A3.** Schematic diagram of temperature coefficients ($Q_{10}$) and post-acclimation $Q_{10}$ calculations. This figure
was modified from Seebacher et al. (2015). Black line and grey line showed the heart rate of limpets from the warm-
acclimated temperature (24 °C) and the cold-acclimated temperature (20 °C), respectively. $Q_{10}$ values for thermal
sensitivities were calculated from data for limpets kept at an acclimation treatment in which heart rate were measured
at two different temperatures. $Q_{10}$ value for post-acclimation thermal sensitivities was calculated across two
temperature acclimation conditions under the same $p$CO$_2$ condition.


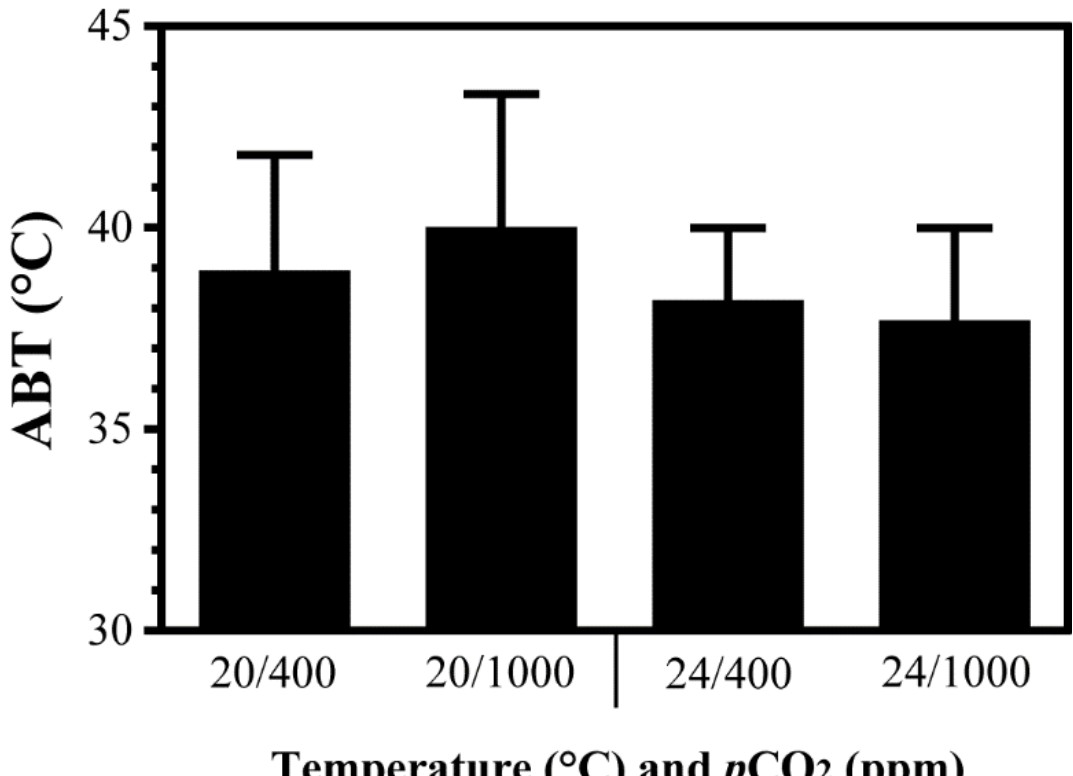



**Figure A4.** Arrhenius Breakpoint Temperature (ABT) of heart rate of limpets acclimated at different temperatures

(20 or 24 °C) and CO2 concentrations (400 or 1000 ppm). After acclimation in different conditions, limpets were

heated continuously from acclimation temperatures to the heart stopped beating. During the heating process, heart

rates were recorded and ABTs were calculated.


**Table A1.** Functions and primers of selected genes of *Cellana* limpet

| Gene name | Gene Symbol | Function | Primers (5'-3') |
|---|---|---|---|
| heat shock cognate 71 kDa protein | *hsc70* | molecular chaperone | F: CCTGAATGTGTCCGCTGTG R: TTCCTGTCTTCCTCGCTGAT |
| heat shock protein 70 | *hsp70* | molecular chaperone | F: CAACACCTTCACGACTTA R: CCACAGCAGATACATTCA |
| beta-actin | *β-actin* | reference gene | F: AGGTATTGCCGACAGAATG R: TTGGAAGGTGGACAGAGA |
| tubulin beta chain | *β-tubulin* | reference gene | F: AGGTGCTGAATTGGTAGAC R: TTGCTGATGAGGAGAGTTC |
| 18S ribosomal RNA | *18s* | reference gene | F: ATAGCCTATATCGGAGTT R: ATGGATACATCAAGGTTAT |


**Table A2.** Inferential statistics for the most likely general additive mixed models (GAMM) of heart rate during
continuous warming of limpet *Cellana toreuma* acclimated at different temperatures (20 and 24 °C) and $pCO_2$ (400
and 1000 ppm)[1]

| Effect | d.f. | *F* | *P*-value |
|---|---|---|---|
| **f(T) for *C. toreuma* from 20 °C and 400 ppm** | **18.46** | **191.2** | **< 0.001** |
| Deviation from *f(T)* for *C. toreuma* from 20 °C and 1000 ppm | 17.2 | 25.018 | < 0.001 |
| Deviation from *f(T)* for *C. toreuma* from 24 °C and 400 ppm | 16.157 | 65.328 | < 0.001 |
| Deviation from *f(T)* for *C. toreuma* from 24 °C and 1000 ppm | 20.194 | 41.634 | < 0.001 |
| **f(T) for *C. toreuma* from 20 °C and 1000 ppm** | **18.75** | **135** | **< 0.001** |
| Deviation from *f(T)* for *C. toreuma* from 24 °C and 400 ppm | 10.502 | 42.441 | < 0.001 |
| Deviation from *f(T)* for *C. toreuma* from 24 °C and 1000 ppm | 19.753 | 40.229 | < 0.001 |
| **f(T) for *C. toreuma* from 24 °C and 400 ppm** | **13.3** | **35.58** | **< 0.001** |
| Deviation from *f(T)* for *C. toreuma* from 24 °C and 1000 ppm | 13.337 | 6.364 | < 0.001 |
| **f(T) for *C. toreuma* from 24 °C and 1000 ppm** | **18.35** | **52.54** | **< 0.001** |

[1]The generalized additive model describes heart rate as a function of temperature, or *f*(T), instead of using a fixed
parameter to describe the effect of temperature. Additional functions were included to describe how heart rates of *C.*
*toreuma* from each treatment deviated from those of *C. toreuma* from 20 °C and 400 ppm.

**Table A3.** Two-way ANOVA to investigate the effects of temperature (20 °C and 24 °C) and $p\text{CO}_2$ (400 ppm and
1000 ppm) on Arrhenius Breakpoint Temperature (ABT) of heart rate and temperature coefficients ($Q_{10}$) on
*Cellana toreuma*

| Source of variation | DF | SS | MS | F | P |
|---|---|---|---|---|---|
| **Two-way ANOVA for ABT** | | | | | |
| Temperature | 1 | 22.580 | 22.580 | 3.375 | 0.075 |
| $p\text{CO}_2$ | 1 | 0.790 | 0.790 | 0.118 | 0.733 |
| Temperature × $p\text{CO}_2$ | 1 | 6.076 | 6.076 | 0.908 | 0.347 |
| Residual | 35 | 234.200 | 6.692 | | |
| **Two-way ANOVA for $Q_{10}$** | | | | | |
| Temperature | 1 | 0.257 | 0.257 | 5.878 | 0.021 |
| $p\text{CO}_2$ | 1 | 0.058 | 0.058 | 1.332 | 0.256 |
| Temperature × $p\text{CO}_2$ | 1 | 0.005 | 0.005 | 0.1135 | 0.738 |
| Residual | 35 | 1.527 | 0.0436 | | |
