# Peer review of "Ocean acidification increases the sensitivity and"

_Biogeosciences, 2017_

## Referee Comment (RC1) · M. Byrne (Referee) · 10 May 2017

The Wang et al ms is an interesting study of the impact of warming and acidification on physiological responses. The main significant effect was seen with the molecular biology – the hsp response. Some analyses of the other parameters measured (eg. heart rate) were equivocal. I suggest reduce the emphasis on the latter and concentrate on the hsp data. Reduce the text on non significant results. I have questions on methods that need to be addressed before a full picture of the outcomes of the work can be assessed.

Introduction

L. 42-45 – Not quite correct there are many studies that show that moderate increase in temperature – within projections – reduces/ameliorates the negative effect of acidification.

At the end of the introduction more contect is need about the region, species and approaches used. Some of this is in the first section of the methods and can be moved here. Also provide some predictions/hypotheses at the end of the introduction. How would you expect the limpets to response with respect to hsp, heart rate, ABT etc.

Methods

Is 7 days a sufficient "acclimation" time – why was this selected. It seems that the limpets were placed directly in treatment – is this a shock? I do not think that with a 7 day experiment much can be said about post-acclimation, (eg. discussion) some justification is needed for this – perhaps there are other studies that have determined this for other limpets.

The sample n=100 per acclimation treatment that is a big sample size, so how many in total $\sim$ 400? How many containers were the limpets in? To use as independent data each limpet would have to be housed in several containers. What was the density of the limpets in each container? These animals have distinct density dependent behaviour – shown in many studies and this may influence outcome. It is not clear to me what was done with the 100's of limpets when only $\sim$ 10 were used for the experimental meaures – perhaps I am missing something?

Show a photo of the artificial rock.

How where the n= 10, n=9-11 limpets selected for hsp and heart rate respectively. Were the latter in separate containers during this measurement?

Use of CV is not mentioned in the stats section – also state why used.

Results

Just provide stats for significant results, so give the ANOVA results for the heart rate and post hoc for the heart rate but not the ABTs. For the latter just give mean and SE, say non significant and cite stats table. Same for the next paragraph.

Fig 2 – why are there no error bars on the post data – best to state why in the legend. Interesting that the hsp data was significant with just n=10 per treatment. Usually n=20 is the minimum.

Discussion

Paragraph 1 can be reduced – some of this is introduction type text Only speak to the significant results and make this clear. State that higher thermal sensitivity to . . .. was indicated by increased heart rate.

It will be good to state what the CVs actually indicate. Overall perhaps for some measures the sample size was too low.

The hsp text could be expanded with regard to the species and methods comparisons. For instance a lot of the work by Tomanek and colleagues involves other intertidal molluscs and on different heights on the shore etc. Are there any other studies of limpets etc.

For the hsp – the sample size may have been too low to discern between constitutive and induced expression.

What studies have used gene expression –vs- protein expression. This might influence the comparisons being made. Just because the gene is expressed we really do not know if the protein is also expressed.

General comments –

L. 21 state 7 days

For a short results section – 6 pages of references seems excessive –

L. 35 Scheffers et al could be deleted

L 46-49 – This is a general sentence – one ref will suffice

L. 96-97 can delete much of this detail (eg falling high tide)

L. 367 – this is a discussion paper – not fully peer review - delete

---

## Referee Comment (RC2) · Anonymous Referee #2 · 31 May 2017

General comments. The manuscript is interesting because it evaluates the physiological effects caused by rising temperatures in limpets acclimated under conditions that combined ocean acidification and temperature rising. The methods, most of all, were well explained, facilitating the understanding of the experiments. However, the limpets were acclimated for a short period of time (7 days) and submitted to different heat shock treatments for a maximum period of 7 h, only once during the whole experiment. No evidence of actual acclimation of these animals was presented (methods for assessing acclimation are discussed by Peck et al. in J. Exp. Biology (2014) 217, 16-22, doi: 10.1242 / jeb.089946). Therefore, contrary to the authors' conclusion the results of these experiments allow predictions of future scenario in a very limited way. The

authors argue about a large variability of the physiological response in the population based on the coefficient of variation of the analyzed parameters. However, this coefficient is derived from a standard deviation that will be reliable if obtained from large population samples, which was not the case (around 10 individuals per treatment). Therefore, this could weaken the argument about the physiological plasticity.

Specific comments Title - The authors obtained evidence that only hsp70 expression was affected in acclimated limpets under HTHC conditions. $CO_2$ level did not affect $Q_{10}$, and the highest temperature decreased $Q_{10}$. Therefore, the ocean acidification affected only hsp70. Then, the title does not specifically reflect the content.

The paragraph between lines 86 and 93 should be in the introduction. The determination of seawater characteristics (lines 112 - 122) should be in a separate item.

The authors should make it clear if the limpets were kept in a chamber with constant $CO_2$ concentration in the air during thermal shock.

On the line 267, the phrase "If only one environmental factor changed (i.e., temperature or $CO_2$) ..." is not sufficiently clear to me.

The discussion about why the expression of hsc70 was not affected by the treatments is insufficient. Why was this protein chosen to analysis? Is it sensitive to temperature rise in other species? Do other factors affect its expression? The discussion needs to be expanded. The conclusion and abstract must be rewritten because an incomplete acclimatization may have occurred and the experiment did not reproduce with reasonable fidelity a future scenario in which the limpets would be exposed to thermal shock.
* * *

---

## Author Comment (AC1) · 2 Jul 2017

Responses to comments of M. Byrne (Referee) "Ocean acidification increases the sensitivity and variability of physiological responses of an intertidal limpet to thermal stress"

The Wang et al ms is an interesting study of the impact of warming and acidification on physiological responses. The main significant effect was seen with the molecular biology – the hsp response. Some analyses of the other parameters measured (eg. heart rate) were equivocal. I suggest reduce the emphasis on the latter and concentrate on the hsp data. Reduce the text on non-significant results. I have questions on methods that need to be addressed before a full picture of the outcomes of the work

can be assessed.

Response: Thanks for your kind and helpful suggestions. Some text about non-significant results were reduced. Otherwise, we expanded hsp discussion in the discussion section. More detailed modifications were providing as follows.

Introduction

Q1: L. 42-45 – Not quite correct there are many studies that show that moderate increase in temperature – within projections – reduces/ameliorates the negative effect of acidification.

Response to Q1: P. 3, L. 43-46. This sentence is changed to: "The interaction between global warming and ocean acidification may not only reduce an organism's resistance to environmental change (Munday et al., 2009), but subsequently affect population dynamics (Fabry et al., 2008; Hoegh-Guldberg et al., 2007; Kroeker et al., 2013; Rodolfo-Metalpa et al., 2011)."

Q2: At the end of the introduction more contect is need about the region, species and approaches used. Some of this is in the first section of the methods and can be moved here. Also provide some predictions/hypotheses at the end of the introduction. How would you expect the limpets to response with respect to hsp, heart rate, ABT etc.

Response to Q2: P. 4-5, L. 72-94. Thanks for your constructive suggestions. The introduction section is reformulated by adding region, species, approaches, and hypotheses, and details are provided as follows. "The limpet C. toreuma is a keystone species on rocky shores in the Western Pacific (Dong et al., 2012) and occupies mid–low intertidal zones. This species is a gonochoric and broadcast spawner, whose embryos develop into planktonic trocophore larvae and later into juvenile veligers before becoming fully grown adults (Ruppert et al., 2004). As a common calcifier inhabiting coastal ecosystem, C. toreuma plays an important ecological role, affecting the community structure of the associated biofilm. Therefore, this species is a key organism

for studying the relationship between physiological response to temperature fluctuation and pH decline in highly variable intertidal zone, with great significance in ecology. Xiamen (118°14'E, 24°42'N) is a representative location in China, which is in a region which is experiencing some of the fastest rates of temperature rise and acidification (reduced pH) globally (Bao and Ren, 2014). The sea surface temperature (SST) in Xiamen coastal area has risen a total of 1 °C since 1960, and is rising at a mean annual rate of 0.02 °C (Yan et al., 2016). The annual pH values of seawater in Xiamen Bay have declined by 0.2 pH units from 1986 to 2012, a trend which is predicted to continue based on simulations (Cai et al., 2016). Here, we investigated the importance of physiological plasticity and variability for C. toreuma to cope with ocean acidification and elevated temperatures by quantifying heart rates (as a proxy of metabolic performance) and expression of genes encoding heat-shock proteins after short-term acclimation in different pCO2 concentrations (400 ppm, 1000 ppm) and temperatures (20 °C, 24 °C). We hypothesize that (1) limpets will increase thermal physiological plasticity under elevated pCO2 and temperatures; (2) limpets acclimatized at different pCO2 concentrations and temperatures can change their heat shock response, and then related energy consumption. This study provides novel information concerning the combined effects of increased temperature and pCO2 on physiological plasticity in intertidal invertebrates, and is important in allowing predications of the ecological impacts of the future environmental changes."

Methods

Q3: Is 7 days a sufficient "acclimation" time – why was this selected. It seems that the limpets were placed directly in treatment – is this a shock? I do not think that with a 7-day experiment much can be said about post-acclimation, (eg. discussion) some justification is needed for this – perhaps there are other studies that have determined this for other limpets.

Response to Q3: Responses were listed separately as follows: (1) It might be proper to describe the 7-day acclimation as a short-term acclimation in the present study. Recent reviews of the literature on the ocean acidification (Doney et al., 2009; Parker et al., 2013) found that the biological responses to acidification between short-term and long-term experiments could be different for benthic invertebrates. We suggest that our study (i.e. short-term acclimation) has its significance for understanding physiological response of organisms to warming and ocean acidification, especially when considering highly variable temperature and $pCO_2$ concentration in the intertidal zone (Cai et al., 2016; Kwiatkowski et al., 2016). Meanwhile, future studies with long-term acclimation (several months) and a larger sample size are recommended in order to validate our findings. (2) Considering that intertidal species under natural conditions can tolerate high variation of temperature and $CO_2$ (Kwiatkowski et al., 2016), we suggest that directly placing the limpets in treatment might not be a strict shock. In addition, in order to avoid the direct shock of treatments, limpets collected in the field were allowed to recover at 20 °C for 3 d with a tidal cycle of approximately 6 h immersion and 6 h emersion in the lab before allocated in treatments. (3) As for the term "post-acclimation", according to Seebacher et al. (2015), the post-acclimation thermal sensitivity is calculated by estimating how much a physiological rate change when animals are allowed to acclimated to different condition (i.e. across chronic acclimation conditions). Since the acclimation is a short-term process in the present study, we suggest that adding the following statement can avoid unnecessary ambiguity. P. 10, L. 193-195: "However, post-acclimation thermal sensitivity should be considered with caution, as the present study was conducted during a short-term acclimation (7 days)."

Q4: The sample n=100 per acclimation treatment that is a big sample size, so how many in total ∼400? How many containers were the limpets in? To use as independent data each limpet would have to be housed in several containers. What was the density of the limpets in each container? These animals have distinct density dependent behavior – shown in many studies and this may influence outcome. It is not clear to me what was done with the 100's of limpets when only ∼10 were used for the experimental measures – perhaps I am missing something?

Response to Q4: Responses to your comments were listed as follows: (1) There were about 100 limpets which were reared in each acclimation treatment. As there were four acclimation treatments, about 400 limpets in total were used for the present study. There were three individuals in a container, and the density was ∼1 limpet per 10 cm2 in each acclimation treatment. As the density in the acclimation treatment is similar to that under field conditions (our field investigation), we thought that the influence of density dependent behavior on the outcome is limited. We suggest that this paragraph could be modified as follows to make it clearer. P. 5-6, L. 98-111: "Samples were collected from Xiamen, and were transported back State Key Laboratory of Marine Environmental Science, Xiamen University, China within 2 h. Limpets were firstly allowed to recover at 20 °C for 3 d with a tidal cycle of approximately 6 h immersion and 6 h emersion. These limpets were randomly allocated into four acclimation treatments and acclimated for 7 d (i.e. short-term acclimation) in different pCO2 concentrations and temperatures (LTLC, 20 °C + 400 ppm, as a control treatment; LTHC, 20 °C + 1000 ppm; HTLC, 24 °C + 400 ppm; HTHC, 24 °C + 1000 ppm) in climate chambers (RXZ280A, Jiangnan Instrument Company, Ningbo, China), which can control the pCO2 concentration. There were about 100 indiv. per acclimation treatment, and the density was ∼ 1 limpet per 10 cm2 in each acclimation treatment. The density in the acclimation treatment is similar to that under field conditions (our field investigation). Control temperature (20 °C) and high temperature (24 °C), respectively, represent the average annual temperature in the collection site and the average global increase (4 °C) predicted for 2100 by the Intergovernmental Panel on Climate Change (IPCC, 2007). Two pCO2 levels, 400 ppm and 1000 ppm, represent the present-day situation and scenarios for 2100 respectively, as projected by IPCC (2007)." (2) In the heat shock experiments, for each acclimation condition, 10 limpets were heated in each designated temperature (26, 30, 34 and 38 °C) and there was a non-heat-stressed group of 10 limpets, so there were 50 individuals in each acclimation treatment. In addition, about 10 individuals were used to test heart rates for each acclimation treatment. Considering that some individuals would die during the acclimation and heat process, ∼

100 individuals were acclimated in each treatment before experiments. The method section about the heat shock experiments was changed to: "After 7-day short-term acclimation, individuals from all four acclimation treatments (n = 10 indiv. per acclimation treatment) were randomly sampled and frozen at -80 °C as non-heated control samples. For each acclimation treatment, 10 limpets were randomly selected from different containers, transferred to an artificial rock (see Fig. A1) and heated at a rate of 6 °C per hour (a natural heating rate, Han et al., 2013) to a designated temperature. There were four designated temperatures (26, 30, 34 and 38 °C). The heat-shock treatments were carried out as described in Denny et al. (2006) (Fig. A2). After achieving the target temperature, the temperature was maintained for the allotted time, and then decreased to acclimated temperatures (20 or 24 °C) at a rate of 6 °C per hour, for a total exposure time of 7 h. After recovery at 20 or 24 °C seawater for 1 h, limpets were immediately collected and stored at -80 °C for gene expression quantification." (P. 7, L. 125-134)

Q5: Show a photo of the artificial rock.

Response to Q5: The photo of the artificial rock (60 cm length $\times$ 30 cm width) was added as shown in Figure A1. Limpets were placed on artificial rock and heated to the designated temperate.

Q6: How where the n= 10, n=9-11 limpets selected for hsp and heart rate respectively. Were the latter in separate containers during this measurement? Use of CV is not mentioned in the stats section – also state why used.

Response to Q6: (1) Limpets were randomly selected from different containers of each acclimation treatment for both gene expression and heart rate experiments. (2) Each limpet was placed in a separate container during the heart rate measurement. (3) The reason why CV is chosen for the present study would be added in the statistical analysis section as follows. P. 10-11, L. 202-206: "The coefficient of variation (CV) of ABT, Q10 and hsc70 mRNA expression at 38 °C were calculated for each acclimation condition. The CV is the variance in a sample divided by the mean of that sample,

providing a method to compare the variation within a sample relative to the mean. It is generally accepted that higher CV demonstrates that there is greater variation among individuals within one treatment than another."

Results

Q7: Just provide stats for significant results, so give the ANOVA results for the heart rate and post hoc for the heart rate but not the ABTs. For the latter just give mean and SE, say non significant and cite stats table. Same for the next paragraph.

Response to Q7: More details about the analysis results would be provided in the results section (P. 11, L. 210-221). "The maximal heart rate was $\sim$ 30 % higher in limpets acclimated to control conditions (20 °C, 400 ppm) than the other treatments (Fig. 1 and Table A3) indicating reduced metabolic performance under high temperatures and $pCO_2$ conditions. The ABTs of limpets ranged from 34.5 °C to 44.2 °C and showed a trend to be reduced for HT treatments (Fig. A4). Temperature (Two-way ANOVA, P = 0.075) and $pCO_2$ (Two-way ANOVA, P = 0.733) both had non-significant effects on ABTs, and there was a non-significant interaction between temperature and $pCO_2$ (Two-way ANOVA, P = 0.347) (Table A4; Fig. A4). Temperature coefficients ($Q_{10}$ rates) were higher for limpets acclimated at 20 °C than at 24 °C (Two-way ANOVA, P = 0.021), but there was no significant difference for acclimation to different $pCO_2$ concentrations and for the interaction between temperature and $pCO_2$ (Two-way ANOVA, P > 0.05) (Table A4; Fig. 2). The post-acclimation thermal sensitivity of limpets acclimated at low $CO_2$ (2.12) was lower than limpets at high $CO_2$ (2.95) (Fig. 2), indicating that the latter are more metabolically sensitive to temperature."

Q8: Fig 2 – why are there no error bars on the post data – best to state why in the legend. Interesting that the hsp data was significant with just n=10 per treatment. Usually n=20 is the minimum.

Response to Q8: (1) According to the formula provided by Seebacher et al. (2015), calculation of post-acclimation $Q_{10}$ is done for the mean response of all individuals

as the same individual are not used at each acclimation temperature. Therefore, no calculation of variation or error is possible. The reason why there are no error bars on the post data would be added in the legend (P. 22, L. 502-504). "The calculation of post-acclimation Q10 is done for the mean response of all individuals as the same individual are not used at each acclimation temperature. Therefore, there was no calculation of variation or error for post-acclimation." (2) Some preliminary researches (e.g. Currie et al., 1999; Dong et al., 2008; Williams et al., 2011; Dong and Williams, 2011; Barshis et al., 2012) were carried out with less than 10 individuals in the heat shock experiments, and showed that such a sample size was reasonable for the hsp gene expression experiment. So we thought that the significance with n=10 was credible.

Discussion

Q9: Paragraph 1 can be reduced – some of this is introduction type text. Only speak to the significant results and make this clear. State that higher thermal sensitivity to .... was indicated by increased heart rate.

Response to Q9: P. 12-13, L. 243-253. Thanks for your useful suggestion. The first paragraph of the discussion section is reduced to: "Higher thermal sensitivity to the pre-dicted future pCO2 (1000 ppm) was indicated by increased heart rate. Post-acclimation thermal sensitivity represents the extent to which ectothermic animals can acclimate to longer-term increases in temperature (several days to weeks) (Seebacher et al., 2015). Thus, the higher thermal sensitivity of limpets acclimated to 1000 ppm indi-cates that the resilience of limpets to thermal stress associated with warming will be compromised under future ocean acidification. This prediction is contrary to the general thought that intertidal ectotherms, such as limpets and other gastropods, will demon-strate high tolerance to thermal stress because they are adapted to an extreme thermal environment. For example, the operative temperatures, from which C. toreuma suffers in the field, frequently exceed 40 °C in summer along Asian coastlines and the limpet can survive at temperatures in excess of 45 °C (Dong et al., 2015). Our data show, however, that ocean acidification will lead to increased sensitivity to changes to future

thermal regimes."

Q10: It will be good to state what the CVs actually indicate. Overall perhaps for some measures the sample size was too low.

Response to Q10: The definition of the coefficients of variation (CV) is stated as follows. "The CV is the variance in a sample divided by the mean of that sample, providing a method to compare the variation within a sample relative to the mean. It is generally accepted that higher CV demonstrates that there is greater variation among individuals within one treatment than another." We aware that our results should be validated by a larger sample size, even though such a sample size (around 10 individuals for each treatment) is reasonable for the hsp gene expression experiment as it has been shown in some researches (e.g. Currie et al., 1999; Dong et al., 2008; Williams et al., 2011; Dong and Williams, 2011; Barshis et al., 2012). Therefore, we recommend that future research should be undertaken with a larger sample size.

Q11: The hsp text could be expanded with regard to the species and methods comparisons. For instance, a lot of the work by Tomanek and colleagues involves other intertidal molluscs and on different heights on the shore etc. Are there any other studies of limpets etc.

Response to Q11: P. 13-14, L. 254-279. The hsp text is expanded by comparing present study with previous researches on intertidal molluscs as follows. "Increased temperature and CO2 increase the sensitivity of heat shock responses to thermal stress. The expression of hsp70 mRNA steadily increased from 20°C to 38°C for individuals across all experimental treatments. However, rates of upregulation of hsp70 mRNA in limpets acclimated at high temperature and high CO2 (HTHC) were significantly higher than those of limpets acclimated at the other three acclimation conditions. As a molecular chaperon, Hsp70 plays crucial roles in maintaining protein stability with the expense of a large amount of energy (Feder and Hofmann, 1999; Tomanek and Sanford, 2003). Usually, the expression of hsp70 of less thermal-tolerant species is

more sensitive to increases in temperature (Dong et al., 2008; Tomanek, 2002). Increasing evidence show that organisms from environments with much stress have a different or increased stress response compared with organisms from environments with less stress. For example, higher intertidal gastropods involved higher heat shock protein expression in response to thermal stress than their lower intertidal counterparts (higher versus lower intertidal: snail Tegula funebralis versus T. brunnea and T. montereyi, Tomanek, 2002, Tomanek and Sanford, 2003; limpet Lottia scabra and L. austrodigitalis versus L. scutum, Dong et al., 2008; limpet C. grata versus C. toreuma, Dong and Williams, 2011). Similar patterns have been observed in hsp70 gene expression for limpet Patella (higher versus lower intertidal: P. rustica versus P. caerulea and P. ulyssiponensis, Prusina et al., 2014). Peck and his colleagues (Peck et al., 2014) found that Atlantic and tropical marine ectotherms are poor in their ability to acclimate physiology to elevated temperature when compared with species from temperate zone. In the present study, the rapid upregulation of hsp70 mRNA in limpets exposed to future conditions (i.e. much stress) potentially represents a high sensitivity of limpets to thermal stress in the face of ocean acidification. Due to the expensive energy consumption during the synthesis and function of hsp70, the more rapid upregulation of hsp70 mRNA in these limpets also indicates more energy was allocated into cellular homeostasis, which then can affect the limpet's growth and reproduction. This change in the metabolic partitioning in individuals could ultimately lead to a decline in fitness and population-level responses."

Q12: For the hsp – the sample size may have been too low to discern between constitutive and induced expression.

Response to Q12: In the present study, the PCR primers (please see Table A2) were used to amplify induced hsp70 gene, which could discern between constitutive and induced expression of hsp70.

Q13: What studies have used gene expression–vs-protein expression. This might influence the comparisons being made. Just because the gene is expressed we really

do not know if the protein is also expressed.

Response to Q13: We assume that the protein is expressed when gene expression occurs for limpets which are heated to designated temperatures, considering that the expression patterns of heat shock protein gene (Zhang et al., 2014; Dong et al., 2014) are similar to the expression patterns of heat shock protein (Tomanek and Somero, 2002; Tomanek, 2002; Tomanek and Sanford, 2003; Dong et al., 2008; Dong and Williams, 2011) for some intertidal gastropods. One of the similar patterns is that both HSP gene expression and protein expression can be rapidly upregulated in respond to heat shock treatment (> 1000 folds more than the control and relatively low temperature shock). Therefore, we suggest that the high-throughput hsp gene expression in respond to heat shock can be translated to heat shock protein in the present study. This speculation needs further experimental evidence in the future study.

General comments –

Q14: L. 21 state 7 days

Response to Q14: P. 2, L. 21. It is changed to: "... (20 °C, 24 °C) regimes in a short-term period (7 days)."

Q15: For a short results section – 6 pages of references seems excessive –

Response to Q15: In the revised manuscript, some redundant references have been deleted.

Q16: L. 35 Scheffers et al could be deleted

Response to Q16: This reference is deleted.

Q17: L 46-49 – This is a general sentence – one ref will suffice

Response to Q17: P. 3, L. 47-49. This sentence is change to: "In the face of a changing environment, organisms have three main options; shift their geographical distribution (Parmesan and Yohe, 2003), develop evolutionary adaptive changes (Hoffmann and

[Figure]

Sgro, 2011), or perish (Fabricius et al., 2011)."

Q18: L. 98-99 can delete much of this detail (eg falling high tide)

Response to Q18: P. 5-6, L98-99. This sentence is reduced to: "Samples were collected from Xiamen, and were transported back State Key Laboratory of Marine Environmental Science, Xiamen University, China within 2 h."

Q19: L. 367 – this is a discussion paper – not fully peer review – delete

Response to Q19: This reference is deleted.

References:

Barshis, D. J., Ladner, J. T., Oliver, T. A., Seneca, F. O., Traylor-Knowles, N., & Palumbi, S. R. (2013). Genomic basis for coral resilience to climate change. Proceedings of the National Academy of Sciences, 110(4), 1387-1392.

Cai, M., Liu, Y., Chen, K., Huang, D., and Yang, S.: Quantitative analysis of anthropogenic influences on coastal water–A new perspective, Ecol. Indic., 67, 673-683, 2016.

Currie, S., Tufts, B. L., & Moyes, C. D. (1999). Influence of bioenergetic stress on heat shock protein gene expression in nucleated red blood cells of fish. American Journal of Physiology-Regulatory, Integrative and Comparative Physiology, 276(4), R990-R996.

Doney, S. C., Fabry, V. J., Feely, R. A., & Kleypas, J. A. (2009). Ocean acidification: the other CO2 problem. Annual review of marine science, 1, 169-192.

Dong, Y. W., Han, G. D., & Huang, X. W. (2014). Stress modulation of cellular metabolic sensors: interaction of stress from temperature and rainfall on the intertidal limpet Cellana toreuma. Molecular ecology, 23(18), 4541-4554.

Dong, Y., Miller, L. P., Sanders, J. G., & Somero, G. N. (2008). Heat-shock protein 70 (Hsp70) expression in four limpets of the genus Lottia: interspecific variation in

constitutive and inducible synthesis correlates with in situ exposure to heat stress. The Biological Bulletin, 215(2), 173-181.

Dong, Y. W., & Williams, G. A. (2011). Variations in cardiac performance and heat shock protein expression to thermal stress in two differently zoned limpets on a tropical rocky shore. Marine biology, 158(6), 1223-1231.

Kwiatkowski, L., Gaylord, B., Hill, T., Hosfelt, J., Kroeker, K. J., Nebuchina, Y., ... & Caldeira, K. (2016). Nighttime dissolution in a temperate coastal ocean ecosystem increases under acidification. Scientific reports, 6.

Parker, L. M., Ross, P. M., O'Connor, W. A., Pörtner, H. O., Scanes, E., & Wright, J. M. (2013). Predicting the response of molluscs to the impact of ocean acidification. Biology, 2(2), 651-692.

Peck, L. S., Morley, S. A., Richard, J., & Clark, M. S.: Acclimation and thermal tolerance in Antarctic marine ectotherms. J. Exp. Biol., 217, 16-22, 2014.

Prusina, I., Sarà, G., De Pirro, M., Dong, Y. W., Han, G. D., Glamuzina, B., & Williams, G. A.: Variations in physiological responses to thermal stress in congeneric limpets in the Mediterranean Sea. J. Exp. Mar. Biol. Ecol., 456, 34-40, 2014.

Seebacher, F., White, C. R., & Franklin, C. E. (2015). Physiological plasticity increases resilience of ectothermic animals to climate change. Nature Climate Change, 5(1), 61-66.

Tomanek, L. (2002). The heat-shock response: its variation, regulation and ecological importance in intertidal gastropods (genus Tegula). Integrative and Comparative Biology, 42(4), 797-807.

Tomanek, L., & Somero, G. N. (2002). Interspecific-and acclimation-induced variation in levels of heat-shock proteins 70 (hsp70) and 90 (hsp90) and heat-shock transcription factor-1 (HSF1) in congeneric marine snails (genus Tegula): implications for regulation of hsp gene expression. Journal of Experimental Biology, 205(5), 677-685.

Tomanek, L., & Sanford, E. (2003). Heat-shock protein 70 (Hsp70) as a biochemical stress indicator: an experimental field test in two congeneric intertidal gastropods (Genus: Tegula). The Biological Bulletin, 205(3), 276-284.

Williams, G. A., De Pirro, M., Cartwright, S., Khangura, K., Ng, W. C., Leung, P. T., & Morritt, D. (2011). Come rain or shine: the combined effects of physical stresses on physiological and protein‐level responses of an intertidal limpet in the monsoonal tropics. Functional Ecology, 25(1), 101-110.

Zhang, S., Han, G. D., & Dong, Y. W. (2014). Temporal patterns of cardiac performance and genes encoding heat shock proteins and metabolic sensors of an intertidal limpet Cellana toreuma during sublethal heat stress. Journal of thermal biology, 41, 31-37.
* * *
[Figure]

[Figure]

**Fig. 1.** The photo of artificial rock (60 cm length × 30 cm width). Limpets were placed on artificial rock and heated to the designated temperate.

---

## Author Comment (AC2) · 2 Jul 2017

Responses to comments of anonymous referee #2 "Ocean acidification increases the sensitivity and variability of physiological responses of an intertidal limpet to thermal stress"

Q1: The methods, most of all, were well explained, facilitating the understanding of the experiments. However, the limpets were acclimated for a short period of time (7 days) and submitted to different heat shock treatments for a maximum period of 7 h, only once during the whole experiment. No evidence of actual acclimation of these animals was presented (methods for assessing acclimation are discussed by Peck et al.

in J. Exp. Biology (2014) 217, 16-22, doi: 10.1242 / jeb.089946).

Therefore, contrary to the authors' conclusion the results of these experiments allow predictions of future scenario in a very limited way. The authors argue about a large variability of the physiological response in the population based on the coefficient of variation of the analyzed parameters. However, this coefficient is derived from a standard deviation that will be reliable if obtained from large population samples, which was not the case (around 10 individuals per treatment). Therefore, this could weaken the argument about the physiological plasticity.

Response to Q1: According to the review by Peck and colleagues (Peck et al., 2014), changes in acute thermal tolerance (upper and lower critical and lethal temperatures, CTmin, CTmax, UTL and LTL) were used to assess the complete acclimation. Though the authors of this review suggested that Antarctic marine invertebrates required 2-5 months to complete whole-animal acclimation, they also pointed out that this conclusion should be noted as the successful acclimation was only observed in a very limited number of species. On the other hand, they suggested that the time needed to acclimate for temperate species is several times lower than that of Antarctic species. In the present study, we did not test the CTmax and thus could not assess the complete acclimation at the whole-animal level in this respect. However, it is also difficult to deny that the short-term acclimation in the present study is not enough for the successful acclimation. As you suggested, we should be careful when making the conclusion that the present results allowed for the prediction of future scenario. We suggest that underlining the short-term acclimation in the conclusion section is important for correctly comprehending the results and conclusions of the present study. There is no doubt that larger sample size can increase the reliability of the CVs. We aware that using the CVs with the sample size (10 individuals per treatment) might weaken the inference about the physiological plasticity. Therefore, in the discussion section we need to state that: "The results about the coefficients of variation need to be interpreted with caution, as the sample size (around 10 limpets per treatment) in the present study may affect the prediction accuracy."

Specific comments

Q2: Title - The authors obtained evidence that only hsp70 expression was affected in acclimated limpets under HTHC conditions. CO2 level did not affect Q10, and the highest temperature decreased Q10. Therefore, the ocean acidification affected only hsp70. Then, the title does not specifically reflect the content.

Response to Q2: Three main findings show the physiological plasticity of limpets acclimated at different conditions. (1) The post-acclimation Q10 of limpets which were acclimated at high pCO2 is much higher than those acclimated at low pCO2, indicating the higher physiological plasticity of limpets to combined environmental stresses. (2) The Coefficients of variation (%) of Arrhenius break temperature (ABT), temperature coefficients (Q10) and hsc70 mRNA expression at 38°C of limpets acclimated at high CO2 are higher than those of the limpets acclimated at low CO2. (3) The rates of upregulation of hsp70 mRNA in limpets acclimated at high temperature and high CO2 (HTHC) were significantly higher than those of limpets acclimated at the other three acclimation conditions. Therefore, we suggest that this title can reflect these three main findings. If the title only presents the significant upregulation of hsp70 mRNA, some other important findings would be lost.

Q3: The paragraph between lines 86 and 93 should be in the introduction. The determination of seawater characteristics (lines 112 - 122) should be in a separate item.

Response to Q3: It is a useful advice and this adjustment would make the manuscript readable.

Q4: The authors should make it clear if the limpets were kept in a chamber with constant CO2 concentration in the air during thermal shock.

Response to Q4: During the thermal shock, the limpets were exposed to air, instead of a chamber with constant CO2 concentration.

Q5: On the line 267, the phrase "If only one environmental factor changed (i.e., tem-

perature or CO2) ..." is not sufficiently clear to me.

Response to Q5: This sentence is rephrased to make it clear. "For limpets acclimated under HTLC and LTHC (i.e., only temperature or CO2 condition changed in comparison to the LTLC treatment), there was significant upregulation of hsc70 mRNA when the heat shock temperatures were beyond 30 °C."

Q6: The discussion about why the expression of hsc70 was not affected by the treatments is insufficient. Why was this protein chosen to analysis? Is it sensitive to temperature rise in other species? Do other factors affect its expression? The discussion needs to be expanded. The conclusion and abstract must be rewritten because an incomplete acclimatization may have occurred and the experiment did not reproduce with reasonable fidelity a future scenario in which the limpets would be exposed to thermal shock.

Response to Q6: (1) Hsc70 is the constitutively expressed protein and is important for the chaperoning function under unstressed conditions, while the Hsp70 is inducible protein and crucial when species suffering acute stress. Basically, Hsc70 and Hsp70 have different expression patterns. However, some studies showed that Hsc70 and Hsp70 have similar response patterns to stress (please see a review by Morris et al. 2013). Also, the response patterns may reflect adaptive strategy to the environment. Therefore, choosing both hsp70 and hsc70 is helpful for us to understand how limpets respond to the heat stress at both constitutive and inducible expression levels.

(2) The expression of hsc70 is the constitutively expressed form and only mildly induced during heat stress. Some studies, however, showed that thermal stress could significantly induce the up-regulation of both hsc70 gene and Hsc70 protein, such as in the killifish Fundulus heteroclitus (Fangue et al. 2006), the shrimp Penaeus monodon (Chuang et al. 2007), and the coral Veretillum cynomorium (Teixeira et al. 2013). The discussion section about hsc70 was expanded as follows. "The expression patterns of hsc70 mRNA were different among limpets at the four acclimation conditions. Hsc70

is constitutively expressed and is a molecular chaperone involved in the in vivo folding and repair of denatured proteins (Dong et al., 2015). Although hsp70 and hsc70 contain similar promoter regions, there are differential expressions to a given stimulus between them (Hansen et al., 1991), which may reflect divergent adaptive strategy to the environment. Some studies showed that thermal stress could significantly induce the up-regulation of both hsc70 gene and Hsc70 protein, such as in the killifish Fundulus heteroclitus (Fangue et al., 2006), the shrimp Penaeus monodon (Chuang et al., 2007), and the coral Veretillum cynomorium (Teixeira et al., 2013). In the present study, the expression of hsc70 mRNA showed no significant difference among different heat-shock temperatures under predicated future environmental conditions (HTHC: 24 °C and 1000 ppm). For limpets acclimated under HTLC and LTHC (i.e., only temperature or CO2 condition changed in comparison with the LTLC treatment), there was significant upregulation of hsc70 mRNA when the heat shock temperatures were beyond 30 °C. These results indicate that expression of hsc70 mRNA is relatively constitutive. That is, the upregulation of hsc70 mRNA in response to heat shock represents an increasing capability for coping with the enhanced protein denaturation and more energy allocated into the somatic maintenance after being exposed to either warming or high CO2 environment. However, the absence of significant upregulation of hsc70 mRNA in limpets acclimated to future conditions (warming and elevated CO2) might be attributed to the very high variation of gene expression at 38°C (CV, 90.36 %). In the context of future conditions, multiple environmental stressors can induce diverse physiological responses among different individuals, which might be an evolutionary adaptation to the harsh environment on the shore."

(3) In addition to heat, other factors like cold, heavy metals, ethanol, toxin, hypoxia and acidosis can also increase the expression of hsc70 (see reviews by Roberts et al., 2010; Liu et al., 2012).

(4) The present study has only investigated the physiological responses of limpets to heat stress after short-term acclimation. Consequently, the abstract and conclusion

sections should be rephrased. The conclusion section was changed to: "In conclusion, the resilience of intertidal limpets to thermal stress is weakened after exposure to predicted future conditions for a short-term acclimation period (7 days). Yet, the combination of elevated temperature and $CO_2$ concentration prompted divergence of physiological and molecular responses. These results suggest that while organisms may be able to protect themselves from the damaging effects of thermal stress in the short-term, changes to multiple environmental conditions may drive population-level responses through physiological responses (e.g. Giomi et al., 2016). Further, the increased variation in responses, and the observation that some individuals were more capable to physiologically cope with the conditions, may be associated with intergenerational adaptation, but this speculation needs further evidence. As the "weaker" individuals are lost, the offspring in the next generation will be better physiologically adapted to warming under high-$CO_2$ conditions. Therefore, while elevated $CO_2$ and the associated ocean acidification decrease the ability of many individuals to respond to thermal stress, it appears that physiological plasticity and variability could be adaptive mechanisms in at least some populations of intertidal organisms. Our research underlined the importance of physiological plasticity and variability for coastal species coping with warming and ocean acidification. However, the present study has only examined the physiological responses of limpets to heat stress after short-term acclimation. Future studies with long-term acclimation (several months) and a larger sample size are therefore recommended in order to validate our findings."

References:

Chuang, K., Ho, S., & Song, Y. (2007). Cloning and expression analysis of heat shock cognate 70 gene promoter in tiger shrimp (Penaeus monodon). Gene, 405(1), 10-18.

Dong, Y., Han, G., Ganmanee, M., & Wang, J. (2015). Latitudinal variability of physiological responses to heat stress of the intertidal limpet Cellana toreuma along the Asian coast. Marine Ecology Progress Series, 107-119.

Fangue, N. A., Hofmeister, M., & Schulte, P. M. (2006). Intraspecific variation in thermal tolerance and heat shock protein gene expression in common killifish, Fundulus heteroclitus. The Journal of Experimental Biology, 209(15), 2859-2872.

Giomi, F., Mandaglio, C., Ganmanee, M., Han, G., Dong, Y., Williams, G. A., & Sara, G. (2016). The importance of thermal history: costs and benefits of heat exposure in a tropical, rocky shore oyster. The Journal of Experimental Biology, 219(5), 686-694.

Liu, T., Daniels, C. K., & Cao, S. (2012). Comprehensive review on the HSC70 functions, interactions with related molecules and involvement in clinical diseases and therapeutic potential. Pharmacology & therapeutics, 136(3), 354-374.

Morris, J. P., Thatje, S., & Hauton, C. (2013). The use of stress-70 proteins in physiology: a re-appraisal. Molecular Ecology, 22(6), 1494-1502.

Peck, L. S., Morley, S. A., Richard, J., & Clark, M. S. (2014). Acclimation and thermal tolerance in Antarctic marine ectotherms. The Journal of Experimental Biology, 217(1), 16-22.

Roberts, R. J., Agius, C., Saliba, C., Bossier, P., & Sung, Y. Y. (2010). Heat shock proteins (chaperones) in fish and shellfish and their potential role in relation to fish health: a review. Journal of fish diseases, 33(10), 789-801.

Teixeira, T., Diniz, M. S., Calado, R., & Rosa, R. (2013). Coral physiological adaptations to air exposure: Heat shock and oxidative stress responses in Veretillum cynomorium. Journal of Experimental Marine Biology and Ecology, 35-41.

---

## Author Response (AR1)

Dear Editor,

Please find below:

     (1)   our point-by-point response to the reviews
     (2)   a list of all relevant changes made in the manuscript
     (3)   a marked-up manuscript version

**(1)**

**Point-by-point response to the reviews**

We thank referees for their positive review of this work. The comments really helped us to improve the manuscript.

For clarity, we keep the review's comments in blue and italic while our response is in black font.

**Reply to comments of M. Byrne (Referee) #1**

*The Wang et al ms is an interesting study of the impact of warming and acidification on physiological responses. The main significant effect was seen with the molecular biology – the hsp response. Some analyses of the other parameters measured (eg. heart rate) were equivocal. I suggest reduce the emphasis on the latter and concentrate on the hsp data. Reduce the text on non-significant results. I have questions on methods that need to be addressed before a full picture of the outcomes of the work can be assessed.*

**Response:** Thanks for your kind and helpful suggestions. Some text about non-significant results were reduced. Otherwise, we expanded *hsp* discussion in the discussion section. More detailed modifications were providing as follows.

*Introduction*
*Q1: L. 42-45 – Not quite correct there are many studies that show that moderate increase in temperature – within projections – reduces/ameliorates the negative effect of acidification.*

**Response to Q1:** P. 3, L. 44-48. This sentence is changed to: "Although ocean acidification can increase the growth of organism in some cases (Gooding et al., 2009), increasing evidence showed that that rising ocean acidity exacerbates global warming, reduces an organism's resistance to environmental change (Munday et al., 2009), but and subsequently affects population dynamics (Fabry et al., 2008; Hoegh-Guldberg et al., 2007; Kroeker et al., 2013; Rodolfo-Metalpa et al., 2011)."

*Q2: At the end of the introduction more contect is need about the region, species and approaches used. Some of this is in the first section of the methods and can be moved here. Also provide some predictions/hypotheses at the end of the introduction. How would you expect the limpets to response with respect to hsp, heart rate, ABT etc.*

**Response to Q2:** P. 4-5, L. 75-98. Thanks for your constructive suggestions. The introduction section is reformulated by adding region, species, approaches, and hypotheses, and details are provided as follows.

"The limpet *C. toreuma* is a keystone species on rocky shores in the Western Pacific (Dong et al., 2012) and occupies mid–low intertidal zones (Morton and Morton 1983). This species is a gonochoric and broadcast spawner, whose embryos develop into planktonic trocophore larvae and later into juvenile veligers before becoming fully grown adults (Ruppert et al., 2004). As a common calcifier inhabiting coastal ecosystem, *C. toreuma* plays an important ecological role in affecting the community structure of the associated biofilm. Therefore, this species is a key organism for studying the relationship between physiological response to thermal stress and ocean acidification in highly variable environment on the shore.

Under the impact of Subtropical High, Xiamen (118°14′ E, 24°42′ N) is one of the hottest areas in China. The coastal seawater of this area is experiencing rapid temperature rise and acidification (Bao and Ren, 2014). The sea surface temperature (SST) in Xiamen coastal area has risen a total of 1 °C since 1960, and is rising at a mean annual rate of 0.02 °C (Yan et al., 2016). The annual pH values of seawater in Xiamen Bay have declined by 0.2 pH units from 1986 to 2012, a trend which is predicted to continue based on simulations (Cai et al., 2016).

Here, we investigated the importance of physiological plasticity and variability for *C. toreuma* to cope with ocean acidification and elevated temperatures by quantifying heart rates (as a proxy of metabolic performance) and expression of genes encoding heat-shock proteins after short-term acclimation in different $pCO_2$ concentrations (400 ppm and 1000 ppm) and temperatures (20 °C and 24 °C). We hypothesize that (1) limpets will increase their thermal sensitivity of metabolism and stress responses under elevated $pCO_2$ and temperatures; (2) short-term acclimation at high temperature and $pCO_2$ will cause higher inter-individual physiological variation. This study provides novel information concerning the combined effects of increased temperature and $pCO_2$ on physiological plasticity in intertidal invertebrates, and is important in allowing predications of the ecological impacts of the future environmental changes."

*Methods*

*Q3: Is 7 days a sufficient "acclimation" time – why was this selected. It seems that the limpets were placed directly in treatment – is this a shock? I do not think that with a 7-day experiment much can be said about post-acclimation, (eg. discussion) some justification is needed for this – perhaps there are other studies that have determined this for other limpets.*

**Response to Q3:** Responses were listed separately as follows:

(1)  It might be proper to describe the 7-day acclimation as a short-term acclimation in the present study. Recent reviews of the literature on the ocean acidification (Doney et al., 2009; Parker et al., 2013) found that the biological responses to acidification between short-term and long-term experiments could be different for benthic invertebrates. We suggest that our study (i.e. short-term acclimation) has its significance for understanding physiological response of organisms to warming and ocean acidification, especially when considering highly variable temperature and $pCO_2$ concentration in the intertidal zone (Cai et al., 2016; Kwiatkowski et al., 2016). Meanwhile, future studies with long-term acclimation (several months) and a larger sample size are recommended in order to validate our findings.

(2)  Considering that intertidal species under natural conditions can tolerate high variation of temperature and $CO_2$ (Kwiatkowski et al., 2016), we suggest that directly placing the limpets in treatment might not be a strict shock. In addition, in order to avoid the direct shock of treatments, limpets collected in the field were allowed to recover at 20 °C for 3 d with a tidal cycle of approximately 6 h immersion and 6 h emersion in the lab before allocated in treatments.

(3)  As for the term "post-acclimation", according to Seebacher et al. (2015), the post-acclimation thermal sensitivity is calculated by estimating how much a physiological rate change when animals are allowed to acclimated to different condition (i.e. across chronic acclimation conditions). Since the acclimation is a short-term process in the present study, we suggest that adding the following statement can avoid unnecessary ambiguity. P. 14, L. 275-276: "Short-term acclimation at elevated temperature and pCO2 can increase physiological sensitivity of limpets against thermal stress."

*Q4: The sample n=100 per acclimation treatment that is a big sample size, so how many in total ~ 400? How many containers were the limpets in? To use as independent data each limpet would have to be housed in several containers. What was the density of the limpets in each container? These animals have distinct density dependent behavior – shown in many studies and this may influence outcome. It is not clear to me what was done with the100's of limpets when only ~10 were used for the experimental measures – perhaps I am missing something?*

**Response to Q4:** Responses to your comments were listed as follows:

(1)   There were about 100 limpets which were reared in each acclimation treatment. As there were four acclimation treatments, about 400 limpets in total were used for the present study. There were three individuals in a container, and the density was ~1 limpet per 10 $cm^2$ in each acclimation treatment. As the density in the acclimation treatment is similar to that when we collected the samples, we thought that the influence of density dependent behavior on the outcome is limited. We suggest that this paragraph could be modified as follows to make it clearer.

    P. 7, L. 98-111: "Samples were collected from Xiamen, and were transported back State Key Laboratory of Marine Environmental Science, Xiamen University, China within 2 h. Limpets were firstly allowed to recover at 20 °C for 3 d with a tidal cycle of approximately 6 h immersion and 6 h emersion. These limpets were randomly allocated into four acclimation treatments and temporally acclimated in different $pCO_2$ concentrations and temperatures (LTLC, 20 °C + 400 ppm, as a control treatment; LTHC, 20 °C + 1000 ppm; HTLC, 24 °C + 400 ppm; HTHC, 24 °C + 1000 ppm) for 7 d in climate chambers (RXZ280A, Jiangnan Instrument Company, Ningbo, China), which can control the $pCO_2$ concentration. There were about 100 indiv. per acclimation treatment, and the density was ~ 1 limpet per 10 cm2 in all acclimation treatments. This density was similar to that when we collected the samples. Control temperature (20 °C) and high temperature (24 °C), respectively, represent the average annual temperature in the collection site and the average global increase (4 °C) predicted for 2100 by the Intergovernmental Panel on Climate Change (IPCC, 2007). Two pCO2 levels, 400 ppm and 1000 ppm, represent the present-day situation and scenarios for 2100 respectively, as projected by IPCC (2007)."

(2)   In the heat shock experiments, for each acclimation condition, 10 limpets were heated in each designated temperature (26, 30, 34 and 38 °C) and there was a non-heat-stressed group of 10 limpets, so there were 50 individuals in each acclimation treatment. In addition, about 10 individuals were used to test heart rates for each acclimation treatment. Considering that some individuals would die during the acclimation and heat process, ~ 100 individuals were acclimated in each treatment before experiments. The method section about the heat shock experiments was changed to: "After 7-day short-term acclimation, individuals from all four acclimation conditions (n = 10 indiv. per acclimation treatment) were randomly sampled and frozen at -80 °C as non-heated control samples. In each acclimation treatment, 40 limpets were randomly selected and were transferred to an artificial rock (see Fig. A1). The rock was heated at a rate of 6 °C per hour (a natural heating rate, Han et al., 2013) to the designated temperatures (26, 30, 34 and 38 °C). The heat-shock treatments were carried out as described in Denny et al. (2006) (Fig. A2). After achieving the target temperature, the temperature was maintained for the allotted time, and then decreased to acclimated temperatures (20 or 24 °C) at a rate of 6 °C per hour, for a total exposure time of 7 h. After recovery at 20 or 24 °C seawater for 1 h, limpets (n = 8-10 indiv. per heat shock temperature at each acclimation condition) were immediately collected and stored at -80 °C for gene expression quantification." (P. 8, L. 147-159)

*Q5: Show a photo of the artificial rock.*

**Response to Q5:** The photo of the artificial rock (60 cm length × 30 cm width) was shown here added as shown in Figure A1. Limpets were placed on artificial rock and heated to the designated temperate.

[Figure]

*Q6: How where the n= 10, n=9-11 limpets selected for hsp and heart rate respectively. Were the latter in separate containers during this measurement? Use of CV is not mentioned in the stats section – also state why used.*

**Response to Q6:**
(1)  Limpets were randomly selected from different containers of each acclimation treatment for both gene expression and heart rate experiments.

(2)  Each limpet was placed in a separate container during the heart rate measurement.

(3)  The reason why CV is chosen for the present study would be added in the statistical analysis section as follows. P. 12, L. 225-229: "The coefficient of variation (CV) of ABT, $Q_{10}$ and *hsc70* mRNA expression at 38 °C were calculated for each acclimation condition. The CV is the variance in a sample divided by the mean of that sample, providing a method to compare the variation within a sample relative to the mean. It is generally accepted that higher CV demonstrates that there is greater variation among individuals within one treatment than another (Reed et al., 2002)."

*Results*

*Q7: Just provide stats for significant results, so give the ANOVA results for the heart rate and post hoc for the heart rate but not the ABTs. For the latter just give mean and SE, say non significant and cite stats table. Same for the next paragraph.*

**Response to Q7:** More details about the analysis results would be provided in the results section (P. 12-13, L. 233-246).

"The maximal heart rate was ~ 30 % higher in limpets acclimated to control conditions (20 °C, 400 ppm) than the other treatments (Fig. 1 and Table A3) indicating reduced metabolic performance under high temperatures and $pCO2$ conditions. The ABTs of limpets ranged from 34.5 °C to 44.2 °C and showed a trend to be reduced for HT treatments (Fig. A4). Temperature (Two-way ANOVA, $F_{1, 35}$ = 3.375, P = 0.075) and $pCO_2$ (Two-way ANOVA, $F_{1, 35}$ = 0.118, P = 0.733) both had non-significant effects on ABTs, and there was a non-significant interaction between temperature and $pCO_2$ (Two-way ANOVA, $F_{1, 35}$ = 0.908, P = 0.347) (Table A4; Fig. A4).

Temperature coefficients ($Q_{10}$ rates) were higher for limpets acclimated at 20 °C than at 24 °C (Two-way ANOVA, $F_{1, 35}$ = 5.878, P = 0.02), but there was no significant difference for acclimation to different $pCO_2$ concentrations (Two-way ANOVA, $F_{1, 35}$ = 1.332, P > 0.05) and for the interaction between temperature and $pCO_2$ (Two-way ANOVA, $F_{1, 35}$ = 0.1135, P > 0.05) (Table A4; Fig. 2). The post-acclimation thermal sensitivity of limpets acclimated at low $CO_2$ (2.12) was lower than limpets at high $CO_2$ (2.95) (Fig. 2), indicating that the latter are more metabolically sensitive to temperature."

*Q8: Fig 2 – why are there no error bars on the post data – best to state why in the legend. Interesting that the hsp data was significant with just n=10 per treatment. Usually n=20 is the minimum.*

**Response to Q8:**
(1) According to the formula provided by Seebacher et al. (2015), calculation of post-acclimation $Q_{10}$ is done for the mean response of all individuals as the same individual are not used at each acclimation temperature. Therefore, no calculation of variation or error is possible. The reason why there are no error bars on the post data would be added in the legend (P. 28, L. 586-588).

"The calculation of post-acclimation $Q_{10}$ is done for the mean response of all individuals as the same individual are not used at each acclimation temperature. Therefore, there was no calculation of variation or error for post-acclimation."

(2) Some preliminary researches (e.g. Currie et al., 1999; Dong et al., 2008; Williams et al., 2011; Dong and Williams, 2011; Barshis et al., 2012) were carried out with less than 10 individuals in the heat shock experiments, and showed that such a sample size was reasonable for the *hsp* gene expression experiment. So we thought that the significance with n=10 was credible.

*Q9: Paragraph 1 can be reduced – some of this is introduction type text. Only speak to the significant results and make this clear. State that higher thermal sensitivity to .... was indicated by increased heart rate.*

**Response to Q9**: P. 12-13, L. 243-253. Thanks for your useful suggestion. The first paragraph of the discussion section is reduced to: "Short-term acclimation at elevated temperature and $p$CO$_2$ can increase physiological sensitivity of limpets against thermal stress. Post-acclimation thermal sensitivity represents the extent to which ectothermic animals can acclimate to longer-term increases in temperature (several days to weeks) (Seebacher et al., 2015). Thus, the higher thermal sensitivity of limpets acclimated to 1000 ppm indicates that the resilience of limpets to thermal stress associated with warming will be compromised under future ocean acidification. This prediction is contrary to the general thought that intertidal ectotherms, such as limpets and other gastropods, will demonstrate high tolerance to thermal stress because they are adapted to an extreme thermal environment. For example, the operative temperatures, from which *C. toreuma* suffers in the field, frequently exceed 40 °C in summer along Asian coastlines and the limpet can survive at temperatures in excess of 45 °C (Dong et al., 2015). Our data show, however, that ocean acidification will lead to increased sensitivity to changes to future thermal regimes."

*Q10: It will be good to state what the CVs actually indicate. Overall perhaps for some measures the sample size was too low.*

**Response to Q10:** The definition of the coefficients of variation (CV) is stated as follows. "The CV is the variance in a sample divided by the mean of that sample, providing a method to compare the variation within a sample relative to the mean. It is generally accepted that higher CV demonstrates that there is greater variation among individuals within one treatment than another."

   We aware that our results should be validated by a larger sample size, even though such a sample size (around 10 individuals for each treatment) is reasonable for the *hsp* gene expression experiment as it has been shown in some researches (e.g. Currie et al., 1999; Dong et al., 2008; Williams et al., 2011; Dong and Williams, 2011; Barshis et al., 2012). Therefore, we recommend that future research should be undertaken with a larger sample size.

*Q11: The hsp text could be expanded with regard to the species and methods comparisons. For instance, a lot of the work by Tomanek and colleagues involves other intertidal molluscs and on different heights on the shore etc. Are there any other studies of limpets etc.*

**Response to Q11**: P. 14-15, L. 287-306. The *hsp* text is expanded by comparing present study with previous researches on intertidal molluscs as follows.

   "Increased temperature and CO$_2$ increase the sensitivity of heat shock responses to thermal stress. The expression of *hsp70* mRNA steadily increased from 20°C to 38°C for individuals across all experimental treatments. However, rates of upregulation of *hsp70* mRNA in limpets acclimated at high temperature and high CO$_2$ (HTHC) were significantly higher than those of limpets acclimated at the other three acclimation conditions. As a molecular chaperon, *Hsp70* plays crucial roles in maintaining protein stability with the expense of a large amount of energy (Feder and Hofmann, 1999; Tomanek and Sanford, 2003). By comparing the expression patterns of Hsp70 of different *Chlorostoma* species (formerly *Tegula*) that have distinct vertical distribution, Tomanek and Somero (1999, 2000) found that there existed interspecific difference in the frequency of the induction of *Hsp70* synthesis and interspecific divergence of the time-course of Hsp70 synthesis.

These studies from genus *Chlorostoma* suggested that species that live higher in the intertidal cost more energy for proteostasis and restore proteostasis to cope with a second consecutive day of high temperatures (Semero et al., 2016). Usually, the expression of *Hsp70* of less thermal-tolerant species is more sensitive to increases in temperature (limpet *Lottia*, Dong et al., 2008; snail *Chlorostoma*, Tomanek, 2002), and the rapid upregulation of *hsp70* mRNA in limpets exposed to future conditions potentially represents a high sensitivity of limpets to thermal stress in the face of ocean acidification. Due to the expensive energy consumption during the synthesis and function of *hsp70*, the more rapid upregulation of *hsp70* mRNA in these limpets also indicates more energy was allocated into cellular homeostasis, which then can affect the limpet's growth and reproduction. This change in the metabolic partitioning in individuals could ultimately lead to a decline in fitness and population-level responses."

*Q12: For the hsp – the sample size may have been too low to discern between constitutive and induced expression.*

**Response to Q12:** In the present study, the PCR primers (please see Table A2) were used to amplify induced *hsp70* gene, which could discern between constitutive and induced expression of *hsp70*.

*Q13: What studies have used gene expression–vs-protein expression. This might influence the comparisons being made. Just because the gene is expressed we really do not know if the protein is also expressed.*

**Response to Q13:** We assume that the protein is expressed when gene expression occurs for limpets which are heated to designated temperatures, considering that the expression patterns of heat shock protein gene (Zhang et al., 2014; Dong et al., 2014) are similar to the expression patterns of heat shock protein (Tomanek and Somero, 2002; Tomanek, 2002; Tomanek and Sanford, 2003; Dong et al., 2008; Dong and Williams, 2011) for some intertidal gastropods. One of the similar patterns is that both HSP gene expression and protein expression can be rapidly upregulated in respond to heat shock treatment (> 1000 folds more than the control and relatively low temperature shock). Therefore, we suggest that the high-throughput hsp gene expression in respond to heat shock can be translated to heat shock protein in the present study. This speculation needs further experimental evidence in the future study.

**General comments –**

*Q14: L. 21 state 7 days*

**Response to Q14:** P. 2, L. 20-21. It is changed to: "… individuals temporally acclimated (7 d) under combinations of different $pCO_2$ (400 ppm and 1000 ppm) and temperature (20 °C and 24 °C) regimes"

*Q15: For a short results section – 6 pages of references seems excessive –*

**Response to Q15:** In the revised manuscript, some redundant references have been deleted.

*Q16: L. 35 Scheffers et al could be deleted*

**Response to Q16:** This reference is deleted.

*Q16: L 46-49 – This is a general sentence – one ref will suffice*

**Response to Q17:** P. 3, L. 49-52. This sentence is change to: "In the face of a changing environment, organisms have three main options; shift their geographical distribution (Parmesan and Yohe, 2003), develop evolutionary adaptive changes (Hoffmann and Sgro, 2011), or perish (Fabricius et al., 2011)."

*Q17: L. 98-99 can delete much of this detail (eg falling high tide)*

**Response to Q18:** P. 7, L98-99. This sentence is reduced to: **"**Samples were collected from Xiamen, and were transported back State Key Laboratory of Marine Environmental Science, Xiamen University, China within 2 h."

*Q18: L. 367 – this is a discussion paper – not fully peer review – delete*

**Response to Q18:** This reference is deleted.

There is no doubt that larger sample size can increase the reliability of the CVs. We aware that using the CVs with the sample size (10 individuals per treatment) might weaken the inference about the physiological plasticity. Therefore, in the discussion section we state that: "However, the results about the coefficients of variation need to be interpreted with caution, as the sample size (around 10 limpets per treatment) in the present study may affect the prediction accuracy."

**Specific comments**

*Q2: Title - The authors obtained evidence that only hsp70 expression was affected in acclimated limpets under HTHC conditions. CO2 level did not affect Q10, and the highest temperature decreased Q10. Therefore, the ocean acidification affected only hsp70. Then, the title does not specifically reflect the content.*

**Response to Q2**: Three main findings show the physiological plasticity of limpets acclimated at different conditions. (1) The post-acclimation $Q_{10}$ of limpets which were acclimated at high $p$CO$_2$ is much higher than those acclimated at low $p$CO$_2$, indicating the higher physiological plasticity of limpets to combined environmental stresses. (2) The Coefficients of variation (%) of Arrhenius break temperature (ABT), temperature coefficients ($Q_{10}$) and *hsc70* mRNA expression at 38°C of limpets acclimated at high $CO_2$ are higher than those of the limpets acclimated at low $CO_2$. (3) The rates of upregulation of *hsp70* mRNA in limpets acclimated at high temperature and high $CO_2$ (HTHC) were significantly higher than those of limpets acclimated at the other three acclimation conditions. Therefore, we suggest that this title can reflect these three main findings. If the title only presents the significant upregulation of *hsp70* mRNA, some other important findings would be lost.

*Q3: The paragraph between lines 86 and 93 should be in the introduction. The determination of seawater characteristics (lines 112 - 122) should be in a separate item.*

**Response to Q3**: It is a useful advice and this adjustment would make the manuscript readable. We have added the paragraph in the introduction.

*Q4: The authors should make it clear if the limpets were kept in a chamber with constant CO2 concentration in the air during thermal shock.*

**Response to Q4**: During the thermal shock, the limpets were exposed to air, instead of a chamber with constant $CO_2$ concentration.

*Q5: On the line 267, the phrase "If only one environmental factor changed (i.e., temperature or CO2) ..." is not sufficiently clear to me.*

**Response to Q5**: This sentence is rephrased to make it clear. "In the present study, for limpets acclimated under HTLC and LTHC (i.e., only temperature or $CO_2$ condition changed in comparison with the LTLC treatment), there was significant upregulation of *hsc70* mRNA when the heat shock temperatures were beyond 30 °C."

*Q6: The discussion about why the expression of hsc70 was not affected by the treatments is insufficient. Why was this protein chosen to analysis? Is it sensitive to temperature rise in other species? Do other factors affect its expression? The discussion needs to be expanded. The conclusion and abstract must be rewritten because an incomplete acclimatization may have occurred and the experiment did not reproduce with reasonable fidelity a future scenario in which the limpets would be exposed to thermal shock.*

**Response to Q6**:
(1) *Hsc70* is the constitutively expressed protein and is important for the chaperoning function under unstressed conditions, while the *Hsp70* is inducible protein and crucial when species suffering acute stress. Basically, *Hsc70* and *Hsp70* have different expression patterns. However, some studies showed that *Hsc70* and *Hsp70* have similar response patterns to stress (please see a review by Morris et al. 2013). Also, the response patterns may reflect adaptive strategy to the environment. Therefore, choosing both *hsp70* and *hsc70* is helpful for us to understand how limpets respond to the heat stress at both constitutive and inducible expression levels.

(2) The expression of *hsc70* is the constitutively expressed form and only mildly induced during heat stress. Some studies, however, showed that thermal stress could significantly induce the up-regulation of both *hsc70* gene and *Hsc70* protein, such as in the killifish *Fundulus heteroclitus* (Fangue et al. 2006), the shrimp *Penaeus monodon* (Chuang et al. 2007), and the coral *Veretillum cynomorium* (Teixeira et al. 2013).

The discussion section about *hsc70* was expanded as follows. "The expression patterns of *hsc70* mRNA were different among limpets at the four acclimation conditions. *Hsc70* is constitutively expressed and is a molecular chaperone involved in the *in vivo* folding and repair of denatured proteins (Dong et al., 2015). Although *hsp70* and *hsc70* contain similar promoter regions, there are differential expressions to a given stimulus between them (Hansen et al., 1991). Some studies showed that thermal stress could significantly induce the up-regulation of both *hsc70* gene and *Hsc70* protein in the killifish *Fundulus heteroclitus* (Fangue et al., 2006), the shrimp *Penaeus monodon* (Chuang et al., 2007), and the coral *Veretillum cynomorium* (Teixeira et al., 2013). In the present study, for limpets acclimated under HTLC and LTHC (i.e., only temperature or $CO_2$ condition changed in comparison with the LTLC treatment), there was significant upregulation of *hsc*70 mRNA when the heat shock temperatures were beyond 30 °C. However, the expression of *hsc70* mRNA showed no significant difference among different heat-shock temperatures under predicated future environmental conditions (HTHC: 24 °C and 1000 ppm). These results indicate that the upregulation of *hsc70* mRNA in response to heat shock represents an increasing capability for coping with the enhanced protein denaturation and more energy allocated into the somatic maintenance after being exposed to either warming or high $CO_2$ environment. The insignificant upregulation of *hsc70* in response to thermal stress indicates that limpets acclimated under HTHC may employ a "preparative defense" strategy (Dong et al., 2008) to maintain high constitutive levels of *hsc70* as a mechanism to copy with unpredictable heat stress. However, the absence of significant upregulation of *hsc70* mRNA in limpets acclimated to future conditions (warming and elevated $CO_2$) might also be attributed to the very high variation of gene expression at 38°C (CV, 90.36 %). In the context of future conditions, multiple environmental stressors can induce diverse physiological responses among different individuals, which might be an evolutionary adaptation to the harsh environment on the shore."

(3)  In addition to heat, other factors like cold, heavy metals, ethanol, toxin, hypoxia and acidosis can also increase the expression of *hsc70* (see reviews by Roberts et al., 2010; Liu et al., 2012).

(4)  The present study has only investigated the physiological responses of limpets to heat stress after short-term acclimation. Consequently, the abstract and conclusion sections should be rephrased.

The conclusion section was changed to: "In conclusion, the resilience of intertidal limpets to thermal stress is weakened after exposure to predicted future conditions for a short-term acclimation period (7 d). Yet, the combination of elevated temperature and $CO_2$ concentration prompted divergence of physiological and molecular responses. These results suggest that while organisms may be able to protect themselves from the damaging effects of thermal stress in the short-term, changes to multiple environmental conditions may drive population-level responses through physiological responses (e.g. Giomi et al., 2016). Further, the increased variation in responses, and the observation that some individuals were more capable to physiologically cope with the conditions, may be associated with intergenerational adaptation, but this speculation needs further evidence. As the "weaker" individuals are lost, the offspring in the next generation will be better physiologically adapted to warming under high-$CO_2$ conditions. Therefore, while elevated $CO_2$ and the associated ocean acidification decrease the ability of many individuals to respond to thermal stress, it appears that physiological plasticity and variability could be adaptive mechanisms in at least some populations of intertidal organisms. Our research underlined the importance of physiological plasticity and variability for coastal species coping with warming and ocean acidification. However, the present study has only examined the physiological responses of limpets to heat stress after short-term acclimation. Future studies with long-term acclimation and a larger sample size are therefore recommended in order to validate our findings."

**(2)**

**A list of all relevant changes made in the manuscript**

Based on the comments of the reviewers, we have intensively discussed the revision of our manuscript. To best possibly address all reviewer's comments, some parts of the manuscript have been updated. Please find below a list of changes that have been made to the manuscript.

- *Abstract*: We underlined the short-term acclimation of the present study and corresponding conclusion was rephrased.

- *Introduction*: The paragraphs about region and species were moved here from methods section. Approaches and hypotheses were added. Some redundant literatures were removed.

- *Material and Methods*: The description about the heating treatment was rephrased. The use of coefficient of variation and the reason why used were stated.

- *Results*: We added some detailed results of the Two-way ANOVA for the analysis of cardiac performance and a table (Table A4) was added in the Appendix.

- *Discussion*: As recommended by the reviewers, the discussion of *hsp70* and *hsc70* should be expanded. The discussion about the responses of heat shock protein was expanded by comparing present study and with previous researches on intertidal molluscs. We mentioned that the conclusion of the present study was made based on the short-term acclimation.

- *Appendix*: A photo of the artificial rock was added in the appendix section. A table of the Two-way ANOVA analysis for the heart rate was provided.

**(3)**

[revised manuscript text omitted]

---

## Referee Report (RR1)

Review of the manuscript entitle "Ocean acidification increases the sensitivity and variability of physiological responses of an intertidal limpet to thermal stress"

This manuscript is an interesting study looking at the impact of warming and ocean acidification on the variability of physiological responses on an intertidal limpet species. The interesting idea in the manuscript is the impact of multiple stressors on the variability of the response. However, since the focus is on inter-individual variability, the sample size should be bigger than 10. 100 individuals were acclimated for each 4 treatments and only about 60 individuals are used for the different experiments.

The material and methods is not clear. 100 individuals are placed in each acclimated treatment. Then 10 are used as a control for genes expression at the end of the acclimation before putting them on the wall. This is not really a control. The 10 individuals for control should have been put on the wall and kept at the control parameters for the same amount of time than the other individuals on the wall. The procedure for the cardiac performance measurement is unclear. How the temperature was heated? There is no control for this procedure. This is also unclear when the limpets are in the water or in the air. Since the individuals are exposed to tidal cycle of 6 hours immersion and 6 hours emersion, is the air temperature of the air the same than the water? Is the wall experiment realized under water or in the air?

The discussion is interesting however, as mentioned by the authors, the sample size limits the scope of the results.

Before publication, the methods have to be clearer so it is easier to interpret the results.

Specific comments:

- Line 37: Widdicombe and Spicer, 2008 missing in the reference list
- Line 43: unclear
- Line 54: Pörtner et al., 2012 missing in the reference list
- Line 64: re-ordered the references
- Line 76: only on the associated biofilm?
- Line 98: when the samples were collected in the field? What was the *in situ* temperature?
- Lines 116-117: unclear
- Line 120: $HSO_4^-$?
- Line 121: Dickson et al. (1990)
- Line 178: Fig. A3
- Line 185: Fig. A3
- Lines 191-192: $R_1$ and $R_2$ were average heart rate?
- Line 278: Fangue et al., 2006 missing in the reference list
- Line 279: Chuang et al., 2007 and Teixeira et al., 2013 missing in the reference list
- Line 456: Year 2010 and not 2012.
- Figure 1: (a) Heart rates of all limpets acclimated at 20°C?
- Lines 554-555: the heart rate of limpets from the warm-acclimated…
- Table A1: change all the umol by μmol and change utam by μatm

---

## Author Response (AR2)

College of Ocean and Earth Sciences          Tel: 86-18659211278

State Key Laboratory of Marine Environmental Science,

Xiamen University, Xiamen, P. R. China          Email: dongyw@xmu.edu.cn

Dec 4th 2017

Dear Carol Robinson,

Thank you and the reviewers so much for your useful comments and suggestions for improving our manuscript, "*Ocean acidification increases the sensitivity and variability of physiological responses of an intertidal limpet to thermal stress*". We have addressed all of the reviewer's comments and feel that they have substantially improved the manuscript.

Please find more details below:
  (1)  our point-by-point response to the reviews
  (2)  a list of all relevant changes made in the manuscript
  (3)  a marked-up manuscript version

Thank you very much for your attention and consideration.

Sincerely,
Yun-wei Dong Ph. D
On behalf of all co-authors.

**(1)**

**Point-by-point response to the reviews**

We thank referees for their positive review of this work. The comments really helped us to improve the manuscript.

For clarity, we keep the review's comments in blue and italic while our response is in black font.

**Reply to comments of M. Byrne (Referee) #1**

*Q1: This manuscript by Wang et al is an interesting study of the impact of climate change stressors on limpets. Several aspects still to be revised. For instance, the significance of doing both the inducible and constitutive forms of HSP needs to be explained in the introduction. Most readers will not appreciate the difference in the two HSPs and what to expect with regard to their expression. What have other studies found? Interestingly the ramping method to assess HSP inducible is different from other more typical 1 hr shock, 1 hr recovery 'heat shock' studies. From a comparative perspective this is important to consider in the discussion.*

**Response to Q1:** Thank you for your constructive suggestions. Responses to your comments were listed as follows:

(1) P. 5, L. 84-92. Generally, expression patterns of inducible and constitutive Hsps are different and the expression is an energy-consuming way of defending thermal stress. Therefore, we underlined the significance of studying both inducible and constitutive *hsps*. We have provided a brief introduction about HSP in the introduction, including its forms and functions in defending thermal stress.

"At the molecular level, expression of heat shock proteins (Hsps) and *hsp* genes is induced above a certain temperature, reaches maximum and finally ceases in response to heat shock (Han et al., 2013; Miller et al., 2009). Upregulation of Hsps and *hsp* genes is an energy-consuming mechanism for defense against thermal stress (Somero et al., 2016). As a commonly used biomarker, the Hsp70 multigenic family includes two proteins with divergent expression patterns (inducible Hsp70 and constitutive Hsc70). Hsp70 significantly increases in expression when animals are exposed to stressors and plays a role in maintaining protein stability (Feder and Hofmann, 1999). Hsc70, which is constitutively expressed and may be mildly induced during stress, takes part in folding and repair of denatured proteins (Dong et al., 2015)."

(2) P. 17, L. 347-357. We have added a paragraph in the discussion to compare expression patterns of *hsp70* under abrupt exposure and gradual exposure (two possible exposure scenarios experienced by intertidal limpets, suggested by Denny et al., 2006) and stated the importance of the present study in predicting how animals will cope with prolonged aerial exposure during low tide.

"Intertidal limpets may experience two sorts of stressful temperature exposures in the field, abrupt or gradual exposure (Denny et al., 2006). The present study showed the upregulation of *hsp70* and *hsc70* expression in *C. toreuma* under gradual exposure. Similar expression patterns have been also observed in Hsp70 under gradual thermal exposure in other intertidal limpets (Dong et al., 2008; Miller et al., 2009). Importantly, the gradual experimental change in thermal environment used here mimics conditions that most intertidal species experience in the field and is important for predicting how animals will resolve prolonged aerial exposure during low tide. Conversely, experimentally simulating abrupt thermal change helps us understand physiological responses to some extreme conditions, such as heat wave (upregulation of *hsp70* in intertidal limpets, Prusina et al., 2014). Therefore, future work combing both abrupt and gradual exposure may offer insight into how intertidal species respond to climate change and extreme weather events in the future."

*Q2: I think the authors may have under sold their work. I do not think that heart beat and both HSPs have been investigated previously as combined response variables in warming-acidification studies. The authors should clearly state what is novel about this study.*

**Response to Q2:** Cardiac responses and heat shock responses are commonly measured physiological responses to climate change (Somero et al., 2016). Although some studies have shown coordinated heart rate and expression of genes encoding to Hsps in response to elevated temperate (Han et al., 2013; Prusina et al., 2014), little is known about the patterns of heart rate and expression of *hsp* genes for coping with warming and ocean acidification. The present study provided insight into combined effects of increased temperature and $pCO_2$ on stress response, energy consumption and physiological plasticity in intertidal invertebrates by measuring both heart rate and expression of *hsp* genes.

P. 4-5, L. 78-95. "Heart rate (HR), as a measure of cardiac activity, is a useful indicator for indicating physiological response to stress in molluscs (Dong and Williams, 2011; Xing et al., 2016). Animals exhibit a stable basal HR under conditions which are not thermally stressful, and HR increases and reaches a peak followed by a sudden decrease with temperature rising (Braby and Somero, 2006; Dong and Williams, 2011). The temperature at which a sharp discontinuity in slope occurs in an Arrhenius plot (i.e. Arrhenius breakpoint temperature, ABT) can represent the limit of metabolic functioning of animals (Nickerson et al., 1989; Somero, 2002). At the molecular level, expression of heat shock proteins (Hsps) and *hsp* genes is induced above a certain temperature, reaches maximum and finally ceases in response to heat shock (Han et al., 2013; Miller et al., 2009). Upregulation of Hsps and *hsp* genes is an energy-consuming mechanism for defense against thermal stress (Somero et al., 2016). As a commonly used biomarker, the Hsp70 multigenic family includes two proteins with divergent expression patterns (inducible Hsp70 and constitutive Hsc70). Hsp70 significantly increases in expression when animals are exposed to stressors and plays a role in maintaining protein stability (Feder and Hofmann, 1999). Hsc70, which is constitutively expressed and may be mildly induced during stress, takes part in folding and repair of denatured proteins (Dong et al., 2015). Some studies have shown coordinated HR and expression of genes encoding to Hsps in response to elevated temperate (Han et al., 2013; Prusina et al., 2014). However, little is known about the patterns of heart rate and expression of *hsp* genes for coping with combined warming and ocean acidification."

P. 6, L. 118-121. "This study provides novel information concerning the combined effects of increased temperature and $pCO_2$ on stress response, energy consumption and physiological plasticity in intertidal invertebrates, potentially providing predications of the ecological impacts of the future environmental changes."

*Q3: I am concerned that the experimental design is psuedoreplicated.*

**Response to Q3:** P. 7, L. 130-137. During the acclimation treatment, all collected limpets were randomly allocated into one of four treatments. In each acclimation treatment, approximately 100

limpets were randomly allocated in ~ 30 containers (3 individuals in each container). All samples were acclimated under the same relative humidity and light intensity conditions with different $p$CO2 concentration and temperature controlled by climate chambers. Therefore, we suggest that the experimental design is not pseudoreplication. In the revised manuscript, we have modified the description about acclimation treatment as follows.

"These limpets were randomly allocated into one of four treatments and temporally acclimated in different $p$CO$_2$ concentrations and temperatures (LTLC, 20 °C + 400 ppm, as a control treatment; LTHC, 20 °C + 1000 ppm; HTLC, 24 °C + 400 ppm; HTHC, 24 °C + 1000 ppm) for 7 d in climate chambers (RXZ280A, Jiangnan Instrument Company, Ningbo, China), which control both the $p$CO2 concentration and temperature under the same relative humidity and light intensity conditions. In each acclimation treatment, approximately 100 limpets were randomly allocated in ~ 30 containers (3 individuals in each container), to simulate filed densities of ~ 1 limpet per 10 cm$^2$."

*Q4: Title and overall interpretations need some consideration (see below)*
*- increasing sensitivity - is this a good or a bad thing?*
*- The variability increases but I do not know what can be said about this because all the limpets per treatment were housed together in one tank and so were competing in a lab environment. This may have influence the outcome.*
*- The limpets may have been collected from different microclimates. This would influence the outcome.*
*In essence - these considerations and potential caveats need to be presented and identify potential limitations of what can be said.*

**Response to Q4:** Thanks for your kind and helpful suggestions. As suggested, we have provided clear descriptions about how samples were collected and how they were treated during the acclimation and heat process in the revised manuscript. We have identified and presented the limitations of the present study. Responses to the different questions are listed separately below:
(1) P. 14, L. 280-290. The increased sensitivity could be negative for the survive of a population. We have modified the first paragraph in the discussion to state the negative outcome of increased sensitivity.

"Short-term acclimation at elevated temperature and $p$CO$_2$ can increase physiological sensitivity of limpets to thermal stress. The higher thermal sensitivity of limpets acclimated to 1000 ppm indicates that the resilience of limpets to thermal stress associated with warming will be compromised under future ocean acidification. This prediction is contrary to the general thought that intertidal ectotherms, such as limpets and other gastropods, will demonstrate high tolerance to thermal stress because they are adapted to an extreme thermal environment. For example, the operative temperatures, which *C. toreuma* suffers in the field, frequently exceed 40 °C in summer along Asian coastlines and the limpet can survive at temperatures in excess of 45 °C (Dong et al., 2015). Our data show, however, that ocean acidification will lead to increased sensitivity to changes to future thermal regimes, indicating a synergistic negative effect. The change in the metabolic partitioning in individuals could ultimately lead to a decline in fitness and population-level responses in the future."

(2) P. 17, L. 343-346. During the acclimation treatment, three individuals were kept in a container, resembling filed densities of ~ 1 limpet per 10 cm$^2$. Despite this, the increased variability could also result from the experiment design. We now acknowledge and presented this limitation as follows: "However, differences among the coefficients of variation need to be interpreted with caution, as multiple factors can cause this type of variation, including the variable environmental history of individuals despite a 7-day acclimation, competition among individuals during the acclimation period, or the sample size (around 10 limpets per treatment)."

(3) Considering high temperature variation from sun-exposed rock surfaces, we only collected samples from shaded rock surfaces. Limpets mainly inhabit the mid-intertidal rocky shores at the collection site. Despite all this, we could not ensure that all samples come from exactly the same microclimate. We identified and presented this limitation in the discussion (please see above; Q4 - (2)).

P. 7, L. 125-128. More details about the sampling are now provided: "Samples were collected from shaded rock surfaces at mid-tidal level in Xiamen on a falling high tide in July (*in situ* temperature: $30.8 \pm 0.8$ °C). The sampling is to ensure that all limpets have similar thermal history, given the possible impacts from microclimate (Dong et al., 2017; Lathlean and Seuront, 2014)."

*Abstract*

*Q5: I am not convinced that the authors have demonstrated physiological plasticity. To identify this, every animal would have to be treated exactly the same, but if they were in the same tank inter-individual interactions may have influence outcome.*

**Response to Q5:** We acclimated the limpets in 'common garden'. After acclimation in different temperatures and $pCO_2$ concentrations, we heat-shocked all the individuals on artificial walls in air separately. There was no direct inter-individual interaction during the heat shock procedure. With this experimental design, we think we can investigate the physiological plasticity. More details on specific changes of experiment designs are provided as follows.

P. 7, L. 125-140. "Samples were collected from shaded rock surfaces at mid-tidal level in Xiamen on a falling high tide in July (*in situ* temperature: $30.8 \pm 0.8$ °C). The sampling is to ensure that all limpets have similar thermal history, given the possible impacts from microclimate (Dong et al., 2017; Lathlean and Seuront, 2014). They were transported to the State Key Laboratory of Marine Environmental Science, Xiamen University, China within 2 h. Limpets were firstly allowed to recover at 20 °C for 3 d with a tidal cycle of approximately 6 h immersion and 6 h emersion. These limpets were randomly allocated into one of four treatments and temporally acclimated in different $pCO_2$ concentrations and temperatures (LTLC, 20 °C + 400 ppm, as a control treatment; LTHC, 20 °C + 1000 ppm; HTLC, 24 °C + 400 ppm; HTHC, 24 °C + 1000 ppm) for 7 d in climate chambers (RXZ280A, Jiangnan Instrument Company, Ningbo, China), which control both the $pCO2$ concentration and temperature under the same relative humidity and light intensity conditions. In each acclimation treatment, approximately 100 limpets were randomly allocated in ~ 30 containers (3 individuals in each container), to simulate filed densities of ~ 1 limpet per 10 cm². Control conditions (20 °C, 400 ppm) represent the average annual temperature and ambient $pCO_2$ (~ 390 ppm) at the collection site, with high temperature (24 °C) and $pCO_2$ (1000 ppm) representing the average global increase (4 °C, 600 ppm) predicted for 2100 by the Intergovernmental Panel on Climate Change (IPCC, 2007)."

P. 8-9, L. 154-167. "After a 7-day acclimation period (crossed $pCO_2 \times$ Temperature treatments, above), the heat-shock treatments were carried out to simulate the gradual temperature exposure of limpets in the filed as described in Denny et al. (2006) (Fig. A1). For each heat-shock treatment, 10 limpets were randomly selected from each of four acclimation conditions (40 indiv. total) and transferred to artificial rocks (Fig. A2), with individuals from LTLC and LTHC on one rock and individuals from HTLC and HTHC on another rock. The artificial rocks were separately placed in 20 °C water baths and 24 °C water baths, and heated at a rate of 6 °C per hour that simulated emersion in the natural condition at the collection site (Han et al., 2013) to the designated temperatures (26, 30, 34 and 38 °C). After achieving the target temperature, the temperature was maintained for the allotted time, and then decreased to the acclimation temperature (20 or 24 °C) at a rate of 6 °C per hour, for a total exposure time of 7 h. Individuals from all four acclimation conditions (n = 10 indiv. per treatment) were randomly selected, transferred to artificial rocks and aerially exposed at 20 or 24 °C for 7 h, as non-heated control samples. After recovery at 20 or 24 °C seawater for 1 h, limpets were immediately collected and stored at -80 °C for gene expression analysis."

*Q6: Individuals from the same population - state from where and that they are intertidal.*

**Response to Q6:** P. 2, L. 16-18. This sentence has been changed to "Here, we evaluate the importance of physiological plasticity for coping with ocean acidification and elevated temperature, and its variability among individuals, of the intertidal limpet *Cellana toreuma* from the same population in Xiamen."

*Q7: Context at lines 107 and 111 need to be added to the abstract - i.e. treatments - what are the control treatments and what based on (e.g. annual mean, temperature at time of collection etc). Where the limpets collected from exactly the same level on the shore and the same aspect wrt to insolation?*

**Response to Q7:** P. 2, L. 18-22. More details about the treatments have been added in the abstract section: "Limpets were collected from shaded mid-intertidal rock surfaces. They were acclimated under combinations of different $p$CO$_2$ concentrations (400 ppm and 1000 ppm, corresponding to pH 7.8 and 8.1) and temperatures (20 °C and 24 °C) in a short-term period (7 days), with the control condition (20 °C and 400 ppm) representing the average annual temperature and present-day $p$CO$_2$ level at the collection site."

*Q8: Let the reader know that both the inducible and constitutive forms of HSP were investigated …and to address for what aspect each are useful/or why used.*

**Response to Q8:** P. 2, L. 22-27. Thanks for your suggestions. Detailed information about the HSP has been provided: "Heart rates (as a proxy for metabolic performance) and genes encoding inducible and constitutive heat-shock proteins (*hsp70* and *hsc70*) at different heat shock temperatures (26, 30, 34 and 38 °C) were measured. Hsp70 and Hsc70 play important roles in protecting cells from heat stresses, but have different expression patterns with Hsp70 significantly increased in expression during stress and Hsc70 constitutively expressed and only mildly induced during stress."

*Q9: I do not understand "better to cope physiologically".*

**Response to Q9:** P. 2, L. 33-34. This sentence has been changed to "… some individuals have higher physiological plasticity to cope with these conditions."

*Q10: Also provide pH levels in the abstract. Because local TA can differ regionally the levels of ppm used are not easy to compare between studies.*

**Response to Q10:** P. 2, L. 20. The pH levels were provided: "… (400 ppm and 1000 ppm, corresponding to pH 8.1 and 7.8) …"

**Introduction**

*Q11: L. 43 - better to replace "acidity" with "decreased pH" - as per Gatusso's paper on how OA should be presented.*

**Response to Q11:** P. 3, L. 49. "acidity" was replaced with "decreased pH".

*Q12: L. 56 replace "climate change" with "increased temperate" so the reader understands the focus.*

**Response to Q12:** P. 4, L. 63. "climate change" was replaced with "increased temperature".

*Q13: L. 61 - correct the Gibson reference - this is the editor of the volume not the author of the paper!*

**Response to Q13:** P. 4, L. 68. The reference has been modified and the sentence was changed to: "… organisms to warming (Byrne and Przeslawski, 2013; Byrne, 2011; Kroeker et al., 2013), …"

*Q14: The main response variables are heart rate, hspc and hspi. All of these and their use as indicators should be introduced. It will suffice to say that heart beat is often used with molluscs in stress studies (REF). Does an increase in heart rate indicate the animal is coping or is in stress - i.e. what does heart rate measures indicate (with refs). Similarly, the two HSP markers need to be introduced. How do these differ? Why do both? Cite a few previous studies. One of the authors, Dr Dong has done great research on HSPs and so will be able to address this. Is this the first study to combine these physiological markers with heart rate? Here the authors can clearly state what is novel about this study.*

**Response to Q14:** Thank you for your constructive suggestions. We have added a paragraph to introduce heart rate and HSP markers (P. 4-5, L. 78-95), and stated the novelty of the present study (P. 6, L. 118-121.). Please see above (Q2) for more details of these changes in the revised manuscript.

*Q15: L. 85 how did the authors measure plasticity - was this based on CV?? If so state this here and justify with refs>*

**Response to Q15:** P. 6, L. 110-112. In the present study, the plasticity was measured based on post-acclimation temperature sensitivity. This sentence was changed to: "Here, we investigated the importance of physiological plasticity (based on the measurement of post-acclimation temperature sensitivity; see Seebacher et al., 2015) and variability (based on coefficient of variation) for *C. toreuma* …"

**Methods**

*Q16: The animals are from fluctuating habitats (e.g. L. 51,77) but the treatments are static. I understand that it is difficult to mimic intertidal flux conditions in the lab, but the fact that the experimental conditions used do not reflect the natural conditions needs to be acknowledged and*

*some justification of the approach stated - perhaps add 'potential' insights?*

**Response to Q16:** P. 17-18, L. 357-359. Thanks for your useful suggestions. We acknowledged that it is necessary to state the experimental conditions used do not reflect fluctuant conditions in natural environment. We have added a sentence to mention this fact in the final paragraph in the discussion.

"Further, since our findings are based on static experimental conditions, the results should be treated with caution when we predict organism's response to future climate change in the highly variable natural environment."

*Q17: We are told what the habitat temperature is. What is the range of the habitat pCO2?*

**Response to Q17:** P. 7, L. 137-138. The habitat $pCO_2$ was provided in the revised manuscript: "… represent the average annual temperature and ambient $pCO_2$ (~ 390 ppm) at the collection site, …"

*Q18: L. 64 - microclimate is mentioned - indeed this is very important. For instance, the Lathlean papers (infrared studies) and others show that depending on aspect of the habitat rocks to the sun and even rock colours, that limpets and other intertidal invertebrates have very different thermal environments. The reader needs to be assured that the experimental animals had a similar thermal history and cite some of these studies.*

**Response to Q18:** P. 7, L. 125-128. Thank you for your useful suggestions. All samples were collected from shaded rock surfaces at mid-intertidal level at the collection site. Therefore, we suggested that all collected limpets had similar microclimate and thermal history. In the revised manuscript, detailed information about the sampling was provided: "Samples were collected from shaded rock surfaces at mid-tidal level in Xiamen on a falling high tide in July (*in situ* temperature: 30.8 ± 0.8 °C). The sampling is to ensure that all limpets have similar thermal history, given the possible impacts from microclimate (Dong et al., 2017; Lathlean and Seuront, 2014)."

*Q19: As I read the design - 100 limpets were maintained in a single container per acclimation treatment. This is pseudoreplication. The authors must acknowledge this. L. 125 - were randomly selected - that helps ... potentially to get around pseudo rep??*

**Response to Q19:** Approximately 100 limpets were reared in each acclimation treatment and they were randomly allocated in ~ 30 containers. There were three individuals in a container, and the density was ~1 limpet per 10 cm$^2$ in each acclimation treatment, similar to that under field conditions (our field investigation) to limit the potential for density-dependent behaviour.

P. 7, L. 135-137. The sentence was changed to: "In each acclimation treatment, approximately 100 limpets were randomly allocated in ~ 30 containers (3 individuals in each container), to simulate filed densities of ~ 1 limpet per 10 cm$^2$."

*Q20: However, the ramping was done on an individual basis.*

**Response to Q20:** P. 8-9, L. 154-167. The procedure of ramping was modified for clarity. Please see above (Q5) for more details of these changes in the revised manuscript.

*Q21: L 177 This is confusing and may be due to English. What does "stands for" mean. It is important that the definition is clear.*

**Response to Q21:** P. 11, L. 213. This sentence was changed to: "Thermal sensitivity is the change in a physiological rate function …".

*Q22: L. 179 replace "is seen" ... perhaps with "was determined"?*
**Response to Q22:** P. 11, L. 215. "is seen" was replaced with "was determined".

*Results*
*Q23: L. 209 ABT - I suggest spell out each use eg.*
**Response to Q23:** P. 12, L. 246-248. The ABTs of all four treatments were provided in the result section: "The ABTs of limpets showed a trend to be reduced for HT treatments (mean ± SD: LTLC, 38.9 ± 2.9 °C; HTLC, 38.2 ± 1.8 °C; LTHC, 40.0 ± 3.3 °C; HTHC, 37.7 ± 2.3 °C) (Fig. A4)."

*Q24: L 208 and 219 "indicating ...." This is discussion text - delete from results.*
**Response to Q24:** They were deleted from results section.

*Discussion*
*Q25: L. 242,254 - what is the bottom line - is sensitivity a good or bad outcome? What is the take home? Move up sentence L. 245 and 271-272. We need to understand what does change in sensitivity mean for the prospects of the limpets in the future.*
**Response to Q25:** P. 14, L. 280-290. Thank you for your constructive suggestions. The bottom line is that the increased sensitivity and stress response of limpets under future conditions could be a negative response, especially for the survival of a population. We have modified the first paragraph in the discussion section:

"Short-term acclimation at elevated temperature and $pCO_2$ can increase physiological sensitivity of limpets to thermal stress. The higher thermal sensitivity of limpets acclimated to 1000 ppm indicates that the resilience of limpets to thermal stress associated with warming will be compromised under future ocean acidification. This prediction is contrary to the general thought that intertidal ectotherms, such as limpets and other gastropods, will demonstrate high tolerance to thermal stress because they are adapted to an extreme thermal environment. For example, the operative temperatures, which *C. toreuma* suffers in the field, frequently exceed 40 °C in summer along Asian coastlines and the limpet can survive at temperatures in excess of 45 °C (Dong et al., 2015). Our data show, however, that ocean acidification will lead to increased sensitivity to changes to future thermal regimes, indicating a synergistic negative effect. The change in the metabolic partitioning in individuals could ultimately lead to a decline in fitness and population-level responses in the future."

*Q26: L. 251 - Does this indicate a synergistic negative effect?*
**Response to Q26:** P. 14, L.288-289. The increased sensitivity to thermal stress under the acclimation at elevated temperature and $pCO_2$ indicates that the resilience of limpets to thermal stress is reduced under future ocean acidification, which indicates a synergistic negative effect. This sentence was changed to: "… will lead to increased sensitivity to changes to future thermal regimes, indicating a synergistic negative effect."

*Q27: The HSP text is confusing - let the reader know that one paragraph is on the inducible and the other is on the constitutive forms AND the significance of these - with citations.*

**Response to Q27:** In the discussion section, we focused on discussing the inducible *hsp70* in the second paragraph (P. 14-15, L. 291-309) and constitutive *hsc70* in the third paragraph (P. 15-16, L. 310-331).

P. 14-15, L. 291-309. "Increased temperature and $CO_2$ elevated the sensitivity of heat shock responses to thermal stress. The expression of inducible *hsp70* mRNA steadily increased from 20°C to 38°C for individuals across all experimental treatments. However, rates of upregulation of *hsp70* mRNA in limpets acclimated at high temperature and high $CO_2$ (HTHC) were significantly higher than those of limpets acclimated at the other three acclimation conditions. As a molecular chaperon, Hsp70 protein plays crucial roles in maintaining protein stability with the expense of a large amount of energy (Feder and Hofmann, 1999; Tomanek and Sanford, 2003). By comparing the expression patterns of Hsp70 of different *Chlorostoma* species (formerly *Tegula*) that have distinct vertical distribution, Tomanek and Somero (1999, 2000) found that there existed interspecific difference in the frequency of the induction of Hsp70 synthesis and interspecific divergence of the time-course of Hsp70 synthesis. These studies from genus *Chlorostoma* suggested that species that live higher in the intertidal cost more energy for proteostasis and restore proteostasis to cope with a second consecutive day of high temperatures (Semero et al., 2016). Usually, the expression of Hsp70 of less thermal-tolerant species is more sensitive to increases in temperature (limpet *Lottia*, Dong et al., 2008; snail *Chlorostoma*, Tomanek, 2002), and the rapid upregulation of *hsp70* mRNA in limpets exposed to future conditions potentially represents a high sensitivity of limpets to thermal stress in the face of ocean acidification. Due to the expensive energy consumption during the synthesis and function of *hsp70*, the more rapid upregulation of *hsp70* mRNA in these limpets also indicates more energy was allocated into cellular homeostasis, which then can affect the limpet's growth and reproduction."

P. 15-16, L. 310-331. "The expression patterns of constitutive *hsc70* mRNA were different among limpets acclimated at the four acclimation conditions. Hsc70 is constitutively expressed and is a molecular chaperone involved in the *in vivo* folding and repair of denatured proteins (Dong et al., 2015). Although *hsp70* and *hsc70* contain similar promoter regions, there are differential expressions to a given stimulus between them (Hansen et al., 1991). Some studies showed that thermal stress could significantly induce the up-regulation of both *hsc70* gene and Hsc70 protein in the killifish *Fundulus heteroclitus* (Fangue et al., 2006), the shrimp *Penaeus monodon* (Chuang et al., 2007), and the coral *Veretillum cynomorium* (Teixeira et al., 2013). In the present study, for limpets acclimated under HTLC and LTHC (i.e. only temperature or $CO_2$ condition changed in comparison with the LTLC treatment), there was significant upregulation of *hsc*70 mRNA when the heat shock temperatures were beyond 30 °C. However, the expression of *hsc70* mRNA showed no significant difference among different heat-shock temperatures under predicated future environmental conditions (HTHC: 24 °C and 1000 ppm). These results indicate that the upregulation of *hsc70* mRNA in response to heat shock represents an increasing capability for coping with the enhanced protein denaturation and more energy allocated into the somatic maintenance after being exposed to either warming or high $CO_2$ environment. The insignificant upregulation of *hsc70* in response to thermal stress indicates that limpets acclimated under HTHC may employ a "preparative defense" strategy (Dong et al., 2008) to maintain high constitutive levels of *hsc70* as a mechanism to copy with unpredictable heat stress. However, the absence of significant upregulation of *hsc70* mRNA in limpets acclimated to future conditions (warming and elevated $CO_2$) might also be attributed to the very high variation of gene expression at 38°C (CV, 90.36 %). In the context of future conditions, multiple environmental stressors can induce diverse physiological responses among different individuals, which might be an evolutionary adaptation to the harsh environment on the shore."

*Q28: Para starting L. 295. Could the "variation" (based on CV) interpreted here as "plasticity" be due to variable environmental history not still present despite a 7 day acclimation. This needs to be considered here. A longer acclimation time would be needed. Also plasticity is reversible. Essentially speak to the data and potential caveats.*

**Response to Q28:** As suggested, the variability (based on CV) could be also caused by the variable environmental history in despite of a 7-day acclimation. In the revised manuscript, the caveat for the interpretation of the variability and long-term acclimation to validate our findings was provided as follows:

P. 17, L. 343-346. "However, differences among the coefficients of variation need to be interpreted with caution, as multiple factors can cause this type of variation, including the variable environmental history of individuals despite a 7-day acclimation, competition among individuals during the acclimation period, or the sample size (around 10 limpets per treatment)."

P. 17-18, L. 357-361. "Further, since our findings are based on static experimental conditions, the results should be treated with caution when we predict organism's response to future climate change in the highly variable natural environment. Therefore, future studies with long-term acclimation, larger sample size, and variable treatment conditions are recommended in order to validate our findings."

*Q29: The caveat of flux (natural) -vs- stable (exp) needs to be stated.*

**Response to Q29:** The caveat of natural and experiment conditions has now been clearly stated in combination with the explanation on acclimation time. See the comment above.

**Reply to comments of anonymous referee #3**

*Q1: This manuscript is an interesting study looking at the impact of warming and ocean acidification on the variability of physiological responses on an intertidal limpet species. The interesting idea in the manuscript is the impact of multiple stressors on the variability of the response. However, since the focus is on inter-individual variability, the sample size should be bigger than 10.*

**Response to Q1:** We thank the reviewer for this comment. In combination with the comments from original Referee #2, it has helped to further refine the discussion of our results. While we appreciate that a larger sample size and geographical distribution of samples would be necessary to characterize the variability in responses in a species, in this study we were testing for the response of individuals within a population to altered environmental conditions. The main results that we report are the physiological and molecular responses within the population. In our results, we clearly show that there is a biological response (statistically significant) to the different conditions, meaning that the replication was, by definition, large enough to detect responses. However, we also provide information on the variation in responses as this could be an important part of how populations and species adapt to changing conditions.

P. 16-17, L. 332-346. We have added some clarification on our interpretation of variation as follows: "Variation and plasticity in both physiological and molecular responses to thermal stress are not only important for coping with future environmental change but also underpin evolutionary and adaptive changes through selective pressures (Franks and Hoffmann, 2012). In the present study, the coefficients of variation in physiological responses of limpets acclimated in simulated future conditions, including ABT, $Q_{10}$ and *hsc70* mRNA, were higher than those in the other three acclimation conditions. Crucially, this means that a subset of individuals in our experimental population might be more physiologically pre-adapted to cope with heat shock. Once acclimated to future climate change scenario (warming and ocean acidification), this variation in physiological performance increased, indicating that in a harsher environment the physiological plasticity of some individuals allows them to modify their physiological tolerance limits and increase chances for survival and reproduction (Williams et al., 2008). Under high selective pressure, these individuals would form the basis for future generations while less plastic individuals would be removed from populations. However, differences among the coefficients of variation need to be interpreted with caution, as multiple factors can cause this type of variation, including the variable environmental history of individuals despite a 7-day acclimation, competition among individuals during the acclimation period, or the sample size (around 10 limpets per treatment)."

*Q2: The material and methods is not clear. 100 individuals are placed in each acclimated treatment. Then 10 are used as a control for genes expression at the end of the acclimation before putting them on the wall. This is not really a control. The 10 individuals for control should have been put on the wall and kept at the control parameters for the same amount of time than the other individuals on the wall. The procedure for the cardiac performance measurement is unclear. How the temperature was heated? There is no control for this procedure. This is also unclear when the limpets are in the water or in the air. Since the individuals are exposed to tidal cycle of 6 hours immersion and 6 hours emersion, is the air temperature of the air the same than the water? Is the wall experiment realized under water or in the air?*

**Response to Q2:** Responses to your comments were listed as follows:

(1) P. 8-9, L. 154-167. In the heat shock experiments, for each acclimation condition, 10 limpets were heated in each designated temperature (26, 30, 34 and 38 °C) in addition to a non-heat-stressed (control) group of 10 limpets. All samples experienced a total exposure time of 7 h in air, including the control samples under 20 or 24 °C. The bottom of artificial rock was heated in water bath, and limpets were on the surface of rock and exposed to air. In the revised manuscript, we have modified this paragraph as follows for clarity:

"After a 7-day acclimation period (crossed $pCO_2$ × Temperature treatments, above), the heat-shock treatments were carried out to simulate the gradual temperature exposure of limpets in the filed as described in Denny et al. (2006) (Fig. A1). For each heat-shock treatment, 10 limpets were randomly selected from each of four acclimation conditions (40 indiv. total) and transferred to artificial rocks (Fig. A2), with individuals from LTLC and LTHC on one rock and individuals from HTLC and HTHC on another rock. The artificial rocks were separately placed in 20 °C water baths and 24 °C water baths, and heated at a rate of 6 °C per hour that simulated emersion in the natural condition at the collection site (Han et al., 2013) to the designated temperatures (26, 30, 34 and 38 °C). After achieving the target temperature, the temperature was maintained for the allotted time, and then decreased to the acclimation temperature (20 or 24 °C) at a rate of 6 °C per hour, for a total exposure time of 7 h. Individuals from all four acclimation conditions (n = 10 indiv. per treatment) were randomly selected, transferred to artificial rocks and aerially exposed at 20 or 24 °C for 7 h, as non-heated control samples. After recovery at 20 or 24 °C seawater for 1 h, limpets were immediately collected and stored at -80 °C for gene expression analysis."

(2) P. 9, L. 170-180. In the cardiac performance experiment, one individual was placed into a container, and the container was immersed in water bath and heated at a rate of 6 °C per hour which mimicked emersion in the natural environment. During the whole process of ramping, limpers were in the air. Since limpets exhibited a stable basal heat rate under no-stress condition, we suggested that no control for this procedure is acceptable. Details about the heating procedure were provided as follows:

"The cardiac performance of limpets was recorded during whole heating processes from the acclimated temperature (20 or 24 °C) to the temperature where the heart stopped beating (n = 9-11 indiv. per acclimation treatment). Each limpet was placed in a separate container during the measurement. The containers were immersed in water baths, allowing the temperature in the container to be increased at a rate of 6 °C per hour that simulated emersion in the natural environment. Heart rates were measured using a non-invasive method (Chelazzi et al., 2001; Dong and Williams, 2011). The heartbeat was detected by means of an infrared sensor fixed with Blue-Tac (Bostik, Staffordshire, UK) on the limpet shell at a position above the heart. Variation in the light-dependent current produced by the heartbeat were amplified, filtered and recorded using an infrared signal amplifier (AMP03, Newshift, Leiria, Portugal) and Powerlab AD converter (4/30, ADInstruments, March-Hugstetten, Germany). Data were viewed and analyzed using Lab Chart (version 7.0)."

**Specific comments:**

**Response to Q3:** The reference has been added in the reference list.

P. 25, L. 555-556. "Widdicombe, S. and Spicer, J. I.: Predicting the impact of ocean acidification on benthic biodiversity: what can animal physiology tell us?, J. Exp. Mar. Biol. Ecol., 366, 187-197, 2008."

**Response to Q4:** P. 3, L. 47-52. This sentence has been changed to: "Although ocean acidification can increase the growth of organisms in some cases (e.g. Gooding et al., 2009), there is increasing evidence that decreased pH exacerbates global warming, and interactions of ocean acidification and warming reduce an organism's resistance to environmental change (Munday et al., 2009) and subsequently affect population dynamics (Fabry et al., 2008; Hoegh-Guldberg et al., 2007; Kroeker et al., 2013; Rodolfo-Metalpa et al., 2011)."

**Response to Q5:** The reference has been added in the reference list.
P. 23, L. 512-513. "Pörtner, H. O.: Integrating climate-related stressor effects on marine organisms: unifying principles linking molecule to ecosystem-level changes, Mar. Ecol. Progr. Ser., 470, 273-290, 2012."

**Response to Q6:** P. 3, L. 47. The order of the references has been modified to: "… (Dong et al., 2014; Firth and Williams, 2009)."

**Response to Q7:** P. 5, L. 99-101. Limpets play an important role in food chains of the intertidal zone. This sentence has been changed to: "As a common calcifier inhabiting coastal ecosystem, *C. toreuma* plays an important ecological role in food chains, gazing on biofilm and being an important food source for other species (e.g. crabs, sea birds and sea stars)."

*Q8: - Line 98: when the samples were collected in the field? What was the in situ temperature?*

**Response to Q8:** P. 7, L. 125-126. The samples were collected on a falling high tide in July. The *in situ* temperature on the shaded rock surface was around 30 °C. Details were provided as follows: "Samples were collected from shaded rock surfaces at mid-tidal level in Xiamen on a falling high tide in July (*in situ* temperature: $30.8 \pm 0.8$ °C)."

**Response to Q9:** P. 8, L. 145-148. This sentence has been changed to: "Total dissolved inorganic carbon (DIC) was measured before and after the acclimation in seawater each time using a dissolved inorganic carbon analyzer (As-C3, Apollo SciTech, Colorado, USA), using a Li-Cor® non-dispersive infrared detector (Li-6252) with a precision of 0.1% (Cai, 2003)."

**Response to Q10:** It is $KSO_4^-$, for the value of $KSO_4$ is the dissociation constant for $HSO_4^-$.

**Response to Q11:** P. 8, L. 151. It has been modified to: "… total Boron was set from Millero et al. (2006), Dickson et al. (1990) and Lee et al. (2010) respectively."

*Q12: - Line 178: Fig. A3*
**Response to Q12:** P. 11, L. 214. It has been changed to Fig. A3.

*Q13: - Line 185: Fig. A3*
**Response to Q13:** P. 11, L. 221. It has been modified to Fig. A3.

*Q14: - Lines 191-192: R1 and R2 were average heart rate?*
**Response to Q14:** P. 11-12, L. 226-229. R1 and R2 were the average heart rate. The sentence has been changed to: "In each $CO_2$ concentration (400 ppm or 1000 ppm), the post-acclimation $Q_{10}$ values were calculated using the same equation as shown above, but $R_2$ was the average heart rate of the warm-acclimated limpets at the acclimated temperature ($T_2 = 24$ °C), and $R_1$ was the average heart rate of cold-acclimated limpets at $T_1 = 20$ °C (Fig. A3, modified from Seebacher et al. (2015))."

*Q15: - Line 278: Fangue et al., 2006 missing in the reference list*
**Response to Q15:** The reference has been added in the reference list.
P. 21, L. 446-448. "Fangue, N. A., Hofmeister, M., and Schulte, P. M.: Intraspecific variation in thermal tolerance and heat shock protein gene expression in common killifish, *Fundulus heteroclitus*, J. Exp. Biol., 209, 2859-2872, 2006."

*Q16: - Line 279: Chuang et al., 2007 and Teixeira et al., 2013 missing in the reference list*
**Response to Q16:** The references have been added in the reference list.
P. 20, L. 412-413. "Chuang, K. H., Ho, S. H., and Song, Y. L.: Cloning and expression analysis of heat shock cognate 70 gene promoter in tiger shrimp (*Penaeus monodon*), Gene, 405, 10-18, 2007."
P. 24, L. 541-543. "Teixeira, T., Diniz, M., Calado, R., and Rosa, R.: Coral physiological adaptations to air exposure: heat shock and oxidative stress responses in *Veretillum cynomorium*, J. Exp. Mar. Biol. Ecol., 439, 35-41, 2013.
"

*Q17: - Line 456: Year 2010 and not 2012.*
**Response to Q17:** P. 24, L. 521. It has been modified to 2010.

*Q18: - Figure 1: (a) Heart rates of all limpets acclimated at 20 ℃?*
**Response to Q18:** P. 27, L. 579. It has been changed to: "Figure 1. (a) Heart rates of all limpets acclimated to 20 °C and 400ppm, …"

*Q19: - Lines 554-555: the heart rate of limpets from the warm-acclimated…*
**Response to Q19:** P. 33, L. 629. This sentence has been modified to: "Black line and grey line showed the heart rate of limpets from the warm-acclimated …".

*Q20: - Table A1: change all the umol by μmol and change utam by μatm*
**Response to Q20:** P. 35. All the umol and utam have changed to μmol and μatm, respectively.

**(2)**

**A list of all relevant changes made in the manuscript**

Based on the comments of the reviewers, we have intensively discussed the revision of our manuscript. To best possibly address all reviewer's comments, some parts of the manuscript have been updated. Please find below a list of changes that have been made to the manuscript.

- *Abstract*: We provided basic information on sample collection, pH levels, treatments, and the inducible and constitutive forms of heat shock protein.

- *Introduction*: We added a paragraph to introduce heart rate and heat shock proteins and stated the novelty of the present study.

- *Material and Methods*: The descriptions about sample collection, acclimation and heat-shock treatments, cardiac performance measurement were rephrased.

- *Results*: Average values of ABTs were provided.

- *Discussion*: A paragraph was added to compare expression patterns of *hsp70* in intertidal limpets under abrupt exposure and under gradual exposure. We identified and presented limitations of the present study.

**(3)**

[revised manuscript text omitted]

---

## Author Response (AR3)

College of Ocean and Earth Sciences        Tel: 86-18659211278

State Key Laboratory of Marine Environmental Science,

Xiamen University, Xiamen, P. R. China      Email: dongyw@xmu.edu.cn

April 15th 2018

Dear Dr. Carol Robinson,

Thank you and Prof. Stephen Hawkins so much for your useful comments and suggestions for improving our manuscript, "*Ocean acidification increases the sensitivity and variability of physiological responses of an intertidal limpet to thermal stress*". We have addressed all of the reviewer's comments.

Please find more details below:

    (1)   our point-by-point response to the reviews
    (2)   a marked-up manuscript version

We feel lucky and honored that our revised paper will be judged acceptable for publication in *Biogeosciences* after minor revision.

Thanks for your assistance, I remain.

Sincerely,
Yun-wei Dong Ph. D
On behalf of all co-authors.

**(1) Point-by-point response to the reviews**

We thank the referee for his positive review of this work. The comments really helped us to improve the manuscript.

**Reply to comments of Stephen J Hawkins (Referee) #4**

*Q1: This is a very general readership journal. At first mention in the introduction clearly explain what "constitutive" means (perhaps in a bracket).*

**Response to Q1:** P. 5, L. 93. In the revised manuscript, a concise description has been added to explain "constitutive" at first mention in the introduction section. "… the constitutive Hsc70, which is transcribed continuously …)."

*Q2: L.31 "showed increased sensitivity" is better*

**Response to Q2:** P. 2, L. 32-33. This sentence has been changed to "These results suggested that limpets showed increased sensitivity and stress response …".

*Q3: L. 34 replace "acidic" with "reduced pH" - the pHs are above 7 - therefore alkaline... Introduction*

**Response to Q3:** P. 2, L. 35-36. This sentence has been modified to "While short-term acclimation to reduced pH seawater decreases the ability …".

*Q4: L. 50 "resistance of an organism" (try to avoid apostrophes with the possessive)*

**Response to Q4:** P. 3, L. 52. In the revised manuscript, we avoided using apostrophe with the possessive throughout the text. This sentence has been changed to "… reduce resistance of an organism to environmental change …".

*Q5: L. 53 I would contend these are "responses" not "options". Options implies conscious choice. Sessile and sedentary organisms do not "shift their ranges" - their ranges shift as individuals occur in different places. Highly mobile species may actually shift their ranges but not limpets. This sloppy wording (and I have done it too) permeates the climate change literature - species do not consciously shift their ranges - their ranges shift.*
*Suggested rephrase: Organisms can respond in three ways: exhibit shifts in distributional ranges (…), evolve adaptive changes (), or perish ().*

**Response to Q5:** P. 3, L. 55-57. We have rephrased this sentence according to your suggestion. "In the face of a changing environment, organisms can respond in three ways: exhibit shifts in distributional ranges (Parmesan and Yohe, 2003), develop adaptive changes (Hoffmann and Sgro, 2011), or perish (Fabricius et al., 2011)."

*Q6: L. 80 not a huge amount of space is saved by abbreviating Heart Rate - in full throughout? (I would not like confusion with Human Resources...)*

**Response to Q6:** We used heart rate to replace its abbreviation HR throughout the revised manuscript according to the comment.

*Q7: L. 83 ...... Breakpoint Temperature - start with capitals as a term.*

**Response to Q7:** P. 5, L. 84-85. This sentence has been changed to "… (i.e. Arrhenius Breakpoint Temperature, ABT) …".

*Q8: L. 100 ecosystems*

**Response to Q8:** P. 5, L. 103. It has been changed to "ecosystems".

*Q9: L. 104 rephrase - do you mean "subtropical high pressure systems" ???*

**Response to Q9:** P. 6, L. 107. Subtropical High refers to subtropical high pressure systems. This sentence has been modified to "Under the impact of subtropical high pressure systems, …".

*Q10: L. 107 from what pH to what pH*

**Response to Q10:** P. 6, L. 111-112. We have added the pH values and it was changed to "… have declined by 0.2 pH units from 8.05 in 1986 to 7.85 in 2012".

*Q11: L. 116 "show increased..."*

**Response to Q11:** P. 6, L. 118-119. This sentence has been changed according to suggestions.

*Q12: L. 118 "Our study, by measuring both heart rate and heats hock proteins,..."*

**Response to Q12:** P. 6, L. 121. This sentence has been rephrased according to the comment.

*Q13: L. 126 "ensured" not ensure and delete "is to"*

**Response to Q13:** P. 7, L. 129. We have rephrased this sentence according to the comment.

*Q14: L. 136,155 field not filed and check throughout.*

**Response to Q14:** Modified throughout the text.

*Q15: L. 187 "the foot muscle..."*

**Response to Q15:** P. 10, L. 190. We have modified this sentence.

*Q16: L. 247 - far too much code - in full at first mention for the different treatments to help the reader.*

**Response to Q16:** P. 7, L. 134-137. In the revised manuscript, we have spelt out the codes in full at first mention. "…in different $p$CO$_2$ concentrations and temperatures (LTLC, low temperature and low CO$_2$, 20 °C + 400 ppm, as a control treatment; LTHC, low temperature and high CO$_2$, 20 °C + 1000 ppm; HTLC, high temperature and low CO$_2$, 24 °C + 400 ppm; HTHC, high temperature and high CO$_2$, 24 °C + 1000 ppm) …".

*Q17: L. 276 ...condition: .... (replace comma with colon)*

**Response to Q17:** P. 14, L. 279. This sentence has been modified according to the comment.

*Q18: L. 301 Spend not cost*

**Response to Q18:** P. 15, L. 304. We have modified this sentence according to the comment.

**Response to Q19:** P. 15, L. 305. Modified the spelling error.

**Response to Q20:** P. 17, L. 358. This sentence has been revised according to the comment.

**Response to Q21:** P. 18, L. 361. This sentence has been modified to "…we predict the response of an organism to …".

*Q22: L. Table A1 could go in the main text near the methods - this is important information.*

**Response to Q22:** P. 26. In the revised manuscript, Table A1 has been moved in the main text as Table 1.

*Q23: L. Figure 1A - it would be good to fit individual light lines for each individual limpet as well as the population average - this shows the individual variability and hence plasticity at the heart of the paper.*

**Response to Q23:** P. 28. We have modified Figure 1A to show the individual variability according to the comment.

**(2)**

[revised manuscript text omitted]